# Probing the Geometry of Diffusion Models with the String Method

**Elio Moreau** [1]  **Florentin Coeurdoux** [1]  **Grégoire Ferre** [1]  **Eric Vanden-Eijnden** [1 2]

## Abstract

Understanding the geometry of learned distributions is fundamental to improving and interpreting diffusion models, yet systematic tools for exploring their landscape remain limited. Standard latent-space interpolations fail to respect the structure of the learned distribution, often traversing low-density regions. We introduce a framework based on the string method that computes continuous paths between samples by evolving curves under the learned score function. Operating on pretrained models without retraining, our approach interpolates between three regimes: pure generative transport, which yields continuous sample paths; gradient-dominated dynamics, which recover minimum energy paths (MEPs); and finite-temperature string dynamics, which compute principal curves—self-consistent paths that balance energy and entropy. We demonstrate that the choice of regime matters in practice. For image diffusion models, MEPs contain high-likelihood but unrealistic "cartoon" images, confirming prior observations that likelihood maxima appear unrealistic; principal curves instead yield realistic morphing sequences despite lower likelihood. For protein structure prediction, our method computes transition pathways between metastable conformers directly from models trained on static structures, yielding paths with physically plausible intermediates. Together, these results establish the string method as a principled tool for probing the modal structure of diffusion models—identifying modes, characterizing barriers, and mapping connectivity in complex learned distributions. Code is available at github.com/Diffusion-Strings.

[1]Capital Fund Management, 23 Rue de l'Université, 75007 Paris
[2]Courant Institute of Mathematical Sciences, New York, NY 10012, USA . Correspondence to: Elio Moreau <elio.moreau@ens.psl.eu>, Florentin Coeurdoux <Florentin.Coeurdoux@cfm.com>, Eric Vanden-Eijnden <eve2@cims.nyu.edu>.

*Proceedings of the 43rd International Conference on Machine Learning*, Seoul, South Korea. PMLR 306, 2026. Copyright 2026 by the author(s).

## 1. Introduction

Generative models based on diffusion and flow matching have achieved remarkable success in learning complex data distributions. These models transport individual samples from noise to data, implicitly encoding an energy landscape through learned score functions. Yet this point-wise perspective obscures the global geometry of the learned distribution: its modal structure, the barriers between modes, and the connectivity of the data manifold.

Understanding pathways between samples has broad applications: morphing between configurations, identifying transition states, and revealing mechanisms underlying rare events. In molecular systems, such pathways explain conformational changes and folding; in other domains, they can illuminate how the learned distribution connects distinct modes. However, current generators provide only samples, not the pathways between them. Exploiting the geometric information encoded in the learned landscape requires a principled definition of transition pathways, along with a computational procedure to find them.

We introduce a framework that evolves entire *curves* of samples—strings—rather than individual points. By controlling how these strings interact with the learned score function, we can probe different aspects of the distribution geometry. Figure 1 previews our main finding: the choice of dynamics reveals a fundamental tension between likelihood and realism. Paths that maximize likelihood traverse "cartoon" configurations—simplified, stylized images that the model assigns high probability but that lie outside the typical set. Paths that account for entropy, in contrast, remain within the typical set and produce perceptually natural transitions. This paper makes three main contributions:

1. We adapt the string method from computational chemistry to work with learned score functions, enabling pathway computation from pretrained generative models without explicit energy functions.

2. We show how three choices of dynamics—pure transport, gradient-dominated, and finite-temperature—reveal complementary aspects of the learned landscape.

3. We demonstrate that accounting for entropy is essential for realistic pathways in high dimensions—principal

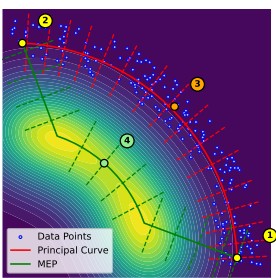
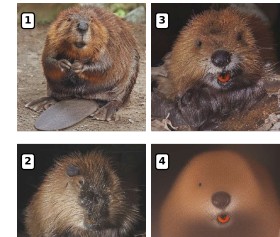

*Figure 1.* **The likelihood-realism paradox.** Top: schematic showing the MEP (green) passing through the high-likelihood region (yellow), while the principal curve (red) stays within the typical set where data concentrates (blue points). Dashed lines indicate Voronoi cells—regions of points closest to each image along the string. Bottom: actual images at numbered locations. Endpoints (1, 2) are identical for both paths; the principal curve intermediate (3) is realistic; the MEP intermediate (4) is cartoonish. Full pathways computed by our method are shown in Figures 4 and 5.

curves traverse the typical set while MEPs do not—and illustrate this on image morphing and protein conformational transitions.

Importantly, our method operates directly on pretrained models without any retraining or fine-tuning. Given only access to the learned velocity field $b_t$ and score $s_t$, we can compute strings in any of the three regimes. This makes the approach immediately applicable to existing models.

### 1.1. Related Work

**Transition path methods.** Computing pathways between metastable states has a long history in computational chemistry. The nudged elastic band method (Henkelman et al., 2000) and string methods (E et al., 2002; Ren et al., 2007; Maragliano et al., 2006) evolve chains of configurations to find minimum (free) energy paths. Extensions like the finite-temperature string method (E et al., 2005) incorporate entropic effects by computing principal curves (Hastie & Stuetzle, 1989), building on concepts from transition path theory (Vanden-Eijnden et al., 2006; E & Vanden-Eijnden, 2010). Other works have similarly sought to compute such pathways by minimizing the Onsager-Machlup action functional (Raja et al., 2025). These methods traditionally require explicit, time-independent energy functions or force fields. Our key contribution is adapting them to time-dependent energies defined implicitly through learned score functions, enabling pathway computation directly from pretrained generative models.

**Generative modeling.** Diffusion models and flows (Ho et al., 2020; Song et al., 2021) learn to reverse a noising process. Stochastic interpolants (Albergo & Vanden-Eijnden, 2023; Albergo et al., 2023), flow matching (Lipman et al., 2023), and rectified flows (Liu et al., 2023) provide uni-

fied frameworks connecting these approaches. Our method builds on this foundation, using the learned velocity and score fields to define string dynamics. Recent work observed that likelihood-maximizing points in diffusion models appear "cartoonish"—perceptually unrealistic despite high probability (Guth et al., 2025; Karczewski et al., 2025). We attribute this paradox to concentration of measure in high dimensions, and resolve it via finite-temperature string dynamics that account for entropy.

**Image morphing.** Interpolating between images in generative models is typically done via linear paths in latent space, but such paths often traverse low-density regions producing unrealistic intermediates. DiffMorpher (Zhang et al., 2024) addresses this through attention interpolation and self-attention guidance; other methods optimize latent trajectories (Wang & Golland, 2023). Our framework differs by grounding interpolation in the geometry of the learned distribution, showing that naive approaches fail by ignoring entropy and that accounting for it yields realistic paths without task-specific modifications.

**Protein conformations.** Recent generative models learn conformational distributions from structural databases: AlphaFlow (Jing et al., 2024) fine-tunes AlphaFold with flow matching, while DiG (Zheng et al., 2023), EigenFold (Jing et al., 2023), ConfDiff (Wang et al., 2024), and FoldingDiff (Wu et al., 2024) train diffusion models on conformational ensembles. These methods sample individual conformations but do not provide transition pathways between them—yet such pathways are essential for understanding protein function, since biological activity often involves conformational changes (Frauenfelder et al., 1991; Henzler-Wildman & Kern, 2007). Our string method complements these approaches by computing pathways directly from the learned score, potentially revealing folding mechanisms and conformational change dynamics from models trained only on static structures.

## 2. Methodology

### 2.1. Generative Models as Score-Based Dynamics

Diffusion and flow-matching models learn to reverse a noising process that transforms data into noise. At the heart of these methods is a time-dependent density $\rho_t(x)$ interpolating between a simple density $\rho_0(x)$ (typically Gaussian) and the data density $\rho_1(x)$. The model learns either a velocity field $b_t(x)$ or a score function $s_t(x) = \nabla \log \rho_t(x)$, which are related through the structure of the interpolation.

Sampling proceeds by integrating a forward-time dynamics.

In its most general form, this takes the shape of an SDE:

$$dx_t = \underbrace{b_t(x_t)\,dt}_{\text{transport}} + \underbrace{\gamma_t^2 s_t(x_t)\,dt}_{\text{score correction}} + \underbrace{\sqrt{2}\gamma_t\,dW_t}_{\text{noise}}, \quad (1)$$

where the volatility $\gamma_t \geq 0$ can be tuned. Setting $\gamma_t = 0$ recovers the deterministic probability flow ODE; positive $\gamma_t$ yields stochastic samplers.

A key observation is that $s_t(x) = -\nabla V_t(x)$ where $V_t(x) = -\log \rho_t(x)$ is an implicit energy landscape. While we cannot evaluate $V_t$ directly, the learned score provides access to its gradient—exactly what the string method requires.

Crucially, the score is estimated via Stein's identity (Gaussian integration by parts), which requires adding noise to the signal. As $t \to 1$, the noise vanishes and this estimator degrades—the learned score becomes unreliable precisely at the data distribution. This is why generative models sample via nonequilibrium transport (pushing noise toward data) rather than Langevin dynamics on $s_1$. For the same reason, we cannot simply evolve strings under $s_1$; instead, we must work dynamically across the full time interval, leveraging reliable scores at intermediate times. More theoretical details on diffusion models can be found in Appendix A.

## 2.2. The String Method: General Formulation

The classical string method (E et al., 2002; Ren et al., 2007) evolves curves under a time-independent potential $V(x)$. Here we generalize to time-dependent velocities $v_t(x)$ arising from diffusion models. For $s \in (0, 1)$, the string evolves according to

$$\dot{\phi}_t(s) = v_t(\phi_t(s)) + \lambda_t(s)\partial_s\phi_t(s), \quad (2)$$

where $\lambda_t(s)$ is a Lagrange multiplier enforcing the constraint $|\partial_s\phi_t(s)| = \text{const}$ in $s$. This constraint prevents bunching or spreading of points—without it, points would cluster at attractors of $v_t$ rather than tracing a path between them. The endpoints $\phi_t(0)$ and $\phi_t(1)$ follow the probability flow ODE $\dot{\phi}_t = b_t(\phi_t)$, which serve as boundary conditions for the string evolution.

**Discrete algorithm.** We discretize the string as $N + 1$ images $\{\phi_t^{(i)}\}_{i=0}^N$. The endpoints $\phi_t^{(0)}$ and $\phi_t^{(N)}$ follow the probability flow; the interior images $i = 1, \ldots, N-1$ evolve via alternating steps (Figure 2):

**Step 1: Evolution.** Move each interior image using $v_t$:

$$\phi_{t+\Delta t}^{(i)} = \phi_t^{(i)} + \Delta t\, v_t(\phi_t^{(i)}), \quad i = 1, \ldots, N-1. \quad (3)$$

**Step 2: Reparametrization.** Redistribute images to restore equal arc-length spacing via linear or cubic spline interpolation (for details see Appendix B.2).

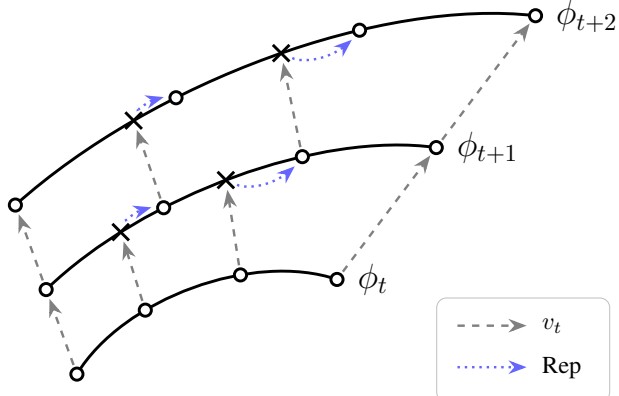

Figure 2. **The string method.** Grey dashed arrows show Step 1 (evolution): each image moves according to $v_t$, landing at positions marked with $\times$. Blue dotted arrows show Step 2 (reparametrization): images are redistributed to restore equal arc-length spacing along the string.

The reparametrization implicitly enforces the $\lambda_t$ constraint. To fully specify the dynamics, we must choose both the velocity field $v_t$ for interior images and an initial string $\{\phi_0^{(i)}\}_{i=0}^N$; both choices are discussed in Section 2.3. For more details on the string method, we refer the reader to Appendix B.

## 2.3. From Transport to Energy to Entropy

What should $v_t$ be? We consider three choices, each motivated by limitations of the previous one.

### 2.3.1. PURE TRANSPORT ($\gamma_t = 0$)

The simplest choice sets $v_t = b_t$, the learned velocity field:

$$\dot{\phi}_t(s) = b_t(\phi_t(s)) + \lambda_t(s)\partial_s\phi_t(s). \quad (4)$$

Starting from a curve of noise samples at $t = 0$, this produces at $t = 1$ a continuous path morphing between generated images. For variance-preserving schedules where $\rho_0 = \mathcal{N}(0, I)$, typical noise samples have norm approximately $\sqrt{d}$. Given two such samples $z_0$ and $z_1$, a natural initialization is

$$\phi_0(s) = z_0\cos(\pi s/2) + z_1\sin(\pi s/2), \quad (5)$$

which traces an approximate geodesic on this sphere; a reparametrization step may be applied to enforce $|\partial_s\phi_0(s)| = \text{const}$ exactly. Alternatively, to morph between two specific data samples $x_A$ and $x_B$, we can integrate the probability flow backward from $t = 1$ to $t = 0$ to obtain $z_0$ and $z_1$ and use these as endpoints—this is the approach taken in our experiments.

Pure transport produces visually appealing morphs, but the resulting path is simply the image of the initial string under

the flow. While individual images remain typical samples of $\rho_t$, the path as a curve is not intrinsically defined—it depends entirely on the initialization. To obtain paths with a principled geometric meaning, we must incorporate the score.

### 2.3.2. MINIMUM ENERGY PATHS ($\gamma_t \gg 1$, $T = 0$)

Adding the score term to the velocity connects the string to the energy landscape $V_t = -\log \rho_t$:

$$v_t = b_t + \gamma_t^2 s_t = b_t - \gamma_t^2 \nabla V_t. \quad (6)$$

In the limit $\gamma_t \to \infty$, the score term dominates and the string relaxes rapidly toward high-density regions of $\rho_t$. Since this relaxation is much faster than the evolution of $V_t$ itself, the string effectively sees a quasi-static energy landscape at each time, recovering the classical string method and computing *minimum energy paths* (MEPs).

> **Definition 2.1** (Minimum Energy Path). A curve $\phi^* : [0, 1] \to \mathbb{R}^d$ is an MEP if the energy gradient (and hence the score) is everywhere tangent to the path:
>
> $$[\nabla V(\phi^*(s))]^\perp = -[s(\phi^*(s))]^\perp = 0, \quad \forall s \in [0, 1],$$
>
> where $[\cdot]^\perp$ denotes the component perpendicular to the tangent $\partial_s \phi^*$.

Since in our setting $V_t = -\log \rho_t$, minimizing energy is equivalent to maximizing likelihood: MEPs are *maximum likelihood paths* connecting two endpoints.

MEPs are geometrically well-defined: they pass through saddle points, identify transition states, and characterize barrier heights. However, in high dimensions, a fundamental problem emerges.

> **The likelihood-realism paradox.** MEPs maximize likelihood along the path, but in high dimensions, high likelihood does not imply high probability mass. Probability mass depends on both the density $\rho(x)$ and the volume of nearby configuration space; the latter is an entropic contribution that dominates in high dimensions. Regions where the product of density and volume is maximized form the *typical set*—where samples actually concentrate.[1] As a result, MEPs traverse modes, which lie outside the typical set.

This explains the "cartoon" phenomenon (Guth et al., 2025; Karczewski et al., 2025): likelihood-maximizing images appear simplified and unrealistic. Our experiments confirm this—as $\gamma$ increases, MEPs traverse higher-likelihood regions with increasingly cartoonish intermediates (Figure 4).

---

[1]For example, the density of a $d$-dimensional Gaussian $\mathcal{N}(0, I)$ is maximal at the origin, but typical samples lie on a sphere of radius $r \approx \sqrt{d}$.

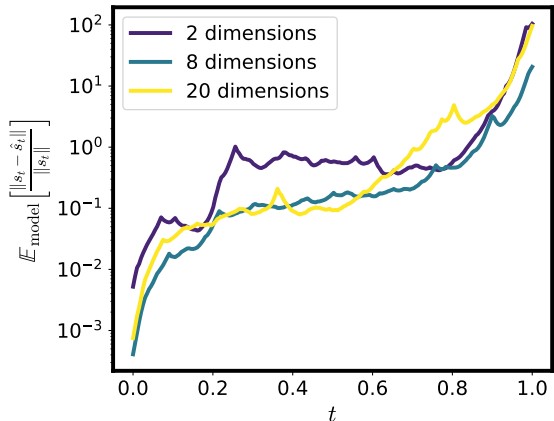

*Figure 3.* **Relative score estimation error:** $\mathbb{E}_{\text{model}}[|s_t - \hat{s}_t|/|s_t|]$ as a function of $t$ for a mixture of Gaussians in various dimensions. The error increases sharply near $t = 1$, motivating the quenching of $\gamma_t$ as $t \to 1$. For details see Appendix C

To find paths connecting samples that remain within the typical set, we must account for entropy. Principal curves, discussed next, achieve this.

### 2.3.3. PRINCIPAL CURVES ($\gamma_t \gg 1$, $T > 0$)

Principal curves were introduced by (Hastie & Stuetzle, 1989) to find structure in unstructured data.

> **Definition 2.2** (Principal Curve). A curve $\phi^* : [0, 1] \to \mathbb{R}^d$ is a principal curve for a distribution $\rho$ if it is self-consistent: each point equals the conditional expectation of points that project onto it:
>
> $$\phi^*(s) = \mathbb{E}_\rho[X \mid s^*(X) = s], \quad \forall s \in [0, 1],$$
>
> where $s^*(X) = \arg\min_{s'} \|X - \phi^*(s')\|$ is the projection of $X$ onto the curve.

In our setting, we take $\rho = \rho_T \propto \rho_1^{1/T}$, where $\rho_1$ is the data distribution and $T \in [0, 1]$ controls the energy-entropy balance, and we fix the endpoints $\phi^*(0) = x_A$ and $\phi^*(1) = x_B$. At $T = 1$, we obtain principal curves for $\rho_1$ connecting the two samples through the typical set; as $T \to 0$, $\rho_T$ concentrates near modes and the principal curve converges to an MEP. Our experiments confirm that increasing $T$ yields increasingly realistic intermediates (Figure 5).

To compute principal curves, we discretize them into $N + 1$ images $\{\phi^{(i)}\}_{i=0}^N$, where the projection regions become Voronoi cells $\mathcal{V}_i = \{x : \|x - \phi^{(i)}\| < \|x - \phi^{(j)}\|$ for all $j \neq i\}$. We associate to each image a walker $x_t^{(i)}$ that samples its Voronoi cell via the full SDE:

$$dx_t^{(i)} = b_t(x_t^{(i)}) \, dt + \gamma_t^2 s_t(x_t^{(i)}) \, dt + \sqrt{2T} \gamma_t \, dW_t, \quad (7)$$

where integration uses timesteps $\Delta t = O(\gamma_t^{-2})$, and we let

each walker drag its string image toward the running average of its position. This effectively defines the velocity $v_t$ of the string. Walkers must also remain closer to their associated string image than to any other; moves violating this constraint are rejected, enforcing the Voronoi restriction. Since $b_t$ and $s_t$ are time-dependent, we impose a separation of timescales via $\gamma_t \gg 1$: the walkers equilibrate within their Voronoi cells much faster than the landscape $V_t$ evolves, so the string tracks an approximate principal curve at each instant.

The complete finite-temperature string method iterates the following steps for $t \in [0, 1]$:

**Step 1: Walker evolution.** Evolve each walker $x_t^{(i)}$ according to (7), rejecting moves that violate the minimum-distance criterion.

**Step 2: String update.** Update each string image via EMA: $\phi_{t+\Delta t}^{(i)} = (1 - \eta)\phi_t^{(i)} + \eta\, x_{t+\Delta t}^{(i)}$, where $\eta \in (0, 1]$ controls the averaging timescale.

**Step 3: Reparametrization.** Redistribute string images to equal arc-length spacing.

*Remark* 2.3. One might ask why not compute principal curves directly on a pregenerated dataset or the original training data. While this could work for curves that remain within high-density regions, we are primarily interested in principal curves connecting metastable states—such as distinct protein conformations or image modes. These transition regions are precisely where data points are scarce. The string method addresses this by using the learned score to sample locally along the curve, even in low-density regions. This also highlights the importance of the boundary conditions in our setting: we seek principal curves connecting two specified endpoints, which differs from Hastie's original formulation where the curve is unconstrained.

### 2.4. Summary and Implementation

Table 1 compares the three regimes in which our framework can operate: pure transport gives appealing but geometrically unmotivated morphs; MEPs provide geometric grounding but fail in high dimensions; principal curves combine geometric meaning with realistic outputs. Algorithm 1 provides the complete procedure for computing strings in any of these regimes.

**Quenching near $t = 1$.** When using $\gamma_t > 0$ (for example to compute MEPs or principal curves), we quench $\gamma_t \to 0$ as $t \to 1$ to avoid amplifying errors in the estimated score near the data distribution (Figure 3). This ensures that interior images arrive accurately at $t = 1$. The endpoints always evolve with pure transport ($\gamma_t = 0$), so they return exactly to $x_A$ and $x_B$.

*Table 1.* Three regimes of string dynamics.

| | Transport $\gamma_t = 0$ | MEP $\gamma_t \gg 1$ $T = 0$ | Principal curve $\gamma_t \gg 1$ $T > 0$ |
|---|---|---|---|
| Geometric | No | Yes | Yes |
| Entropic | No | No | Yes |
| Realistic | Yes | No | Yes |

---

**Algorithm 1** Diffusion String Method

---

**Require:** Data samples $x_A, x_B$; velocity $b_t$; score $s_t$
**Require:** Parameters $\gamma_t, T \in [0, 1]$; initial time $t_0$; number of images $N$; timestep $\Delta t = O(\gamma_t^{-2})$
1: $z_0, z_1 \leftarrow$ integrate $\dot{x}_t = b_t(x_t)$ from $t = 1$ to $t = 0$ starting at $x_A, x_B$
2: Initialize: $\phi_{t_0}^{(i)} = z_0 \cos(\pi i/2N) + z_1 \sin(\pi i/2N)$ for $i = 0, \ldots, N$
3: Reparametrize $\{\phi_{t_0}^{(i)}\}$ to equal arc-length spacing
4: **while** $t < 1$ **do**
5:     **for** $i = 0$ to $N$ **do**
6:         Evolve $\phi_t^{(i)}$ (and walker $x_t^{(i)}$ if $T > 0$) according to chosen regime
7:     **end for**
8:     Reparametrize to equal arc-length spacing
9: **end while**
**Output:** String $\{\phi_1^{(i)}\}_{i=0}^N$ connecting $x_A$ to $x_B$

---

**Computational cost.** The three regimes differ in expense. Pure transport ($\gamma_t = 0$) requires only forward integration of $b_t$ with moderate timesteps. MEPs and principal curves ($\gamma_t \gg 1$) require smaller timesteps $\Delta t = O(\gamma_t^{-2})$ for stability, which dominates the computational cost. In practice, $N = 50$–$70$ images, and for principal curves $\eta = 0.1$–$0.5$, provide a good balance between path resolution and cost. Our focus is on interpretability rather than speed: the method provides a tool for analyzing the geometry of pretrained diffusion models, requiring only access to $b_t$ and $s_t$ without any retraining. Our experiments demonstrate that the approach is practical for realistic applications.

## 3. Experiments

We demonstrate the string method on two domains: ImageNet for the likelihood-realism paradox, and proteins for conformational transitions.

### 3.1. Images: The Three Regimes on ImageNet

We apply the string method to ImageNet ($256 \times 256$) using the SiT-XL-2-256 model (Ma et al., 2024). Images are encoded to a $4 \times 32 \times 32$ latent space via a VAE. Details of the model architecture are given in Appendix D. Additional image pathways are shown in Appendix H.

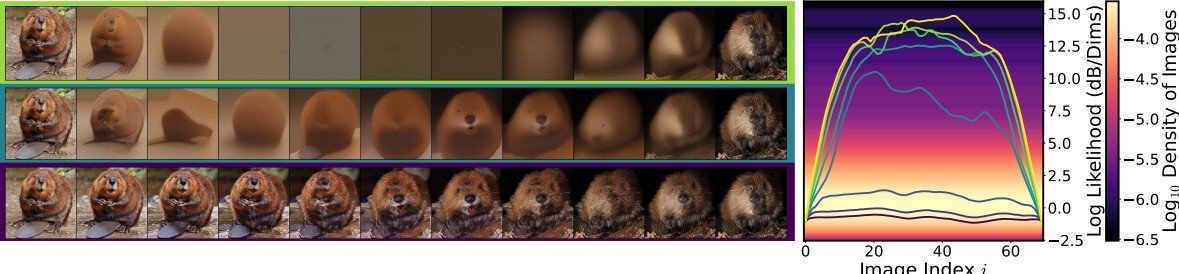

*Figure 4.* **Effect of score weight** $\gamma$. Left: string realizations for $\gamma$ ranging from 15 (top) to 2 (middle) to $10^{-2}$ (bottom) in logarithmic steps (factor of $\sqrt{10}$). Higher $\gamma$ drives paths through abstract, high-likelihood modes; lower $\gamma$ preserves realism. Right: log-likelihood of images along each string (colored curves), overlaid on the likelihood distribution of ImageNet validation images (heatmap; see Figure 6 for details). The intermediate images along the MEP reach likelihoods far exceeding typical images.

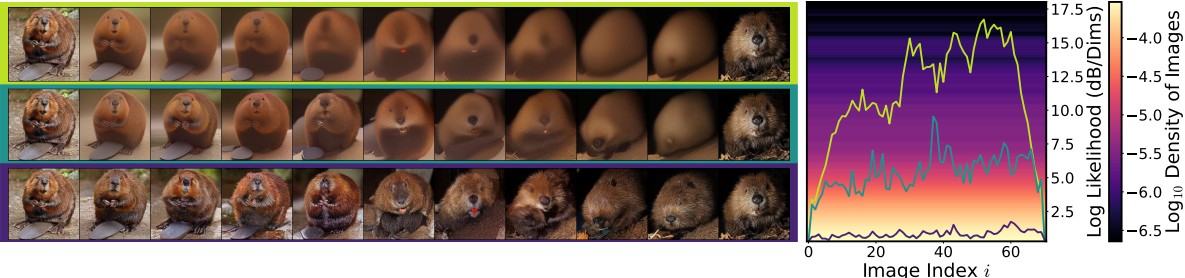

*Figure 5.* **Effect of temperature** $T$. Left: string realizations for $T$ ranging from 0.1 (top) to 0.5 (middle) to 0.9 (bottom). Lower $T$ drives paths through cartoon-like, high-likelihood regions; higher $T$ produces realistic samples. Right: log-likelihood of images along each string (colored curves), overlaid on the likelihood distribution of ImageNet validation images (heatmap; see Figure 6). As $T$ increases, the likelihood of the intermediates images decreases toward typical values, and images become more realistic.

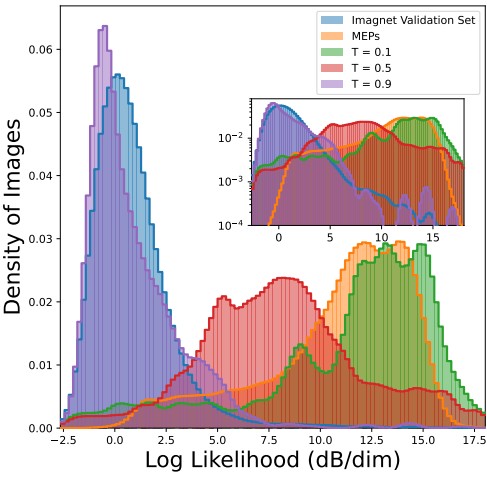

*Figure 6.* **Likelihood distributions.** Histogram of log-likelihoods for ImageNet $256 \times 256$ validation images (blue), compared with images along MEPs (orange) and finite-temperature strings at $T = 0.1$ (green), $T = 0.5$ (red), and $T = 0.9$ (purple), aggregated across multiple strings. MEP intermediates have significantly higher likelihood than real images, confirming they lie outside the typical set. As temperature increases, string images approach the likelihood distribution of real data. Inset: same data on log scale.

**Setup.** Given two images, we: (1) encode into the Gaussian latent space by backward ODE to $t = 0$, (2) initialize a discrete string of $N = 71$ points along a spherical geodesic Eq (5), and (3) evolve the string according to Eq (7) with score weight $\gamma_t = \gamma$ for $0.1 \geq t \geq 0.95$ and $\gamma_t = 0$ otherwise, where $\gamma$ is a tunable parameter described below. This schedule prevents the string from drifting away from the typical set (i.e., the sphere in Gaussian latent space) near $t = 0$, while avoiding regions where the score approximation degrades as $t \rightarrow 1$ (Figure 3).

**Effect of score weight** $\gamma$ **at** $T = 0$. Figure 4 visualizes strings interpolating between two beaver images for varying $\gamma$ and $T = 0$. Full strings are size 71, but we show only of subsample of 11 (1 image every 6).

With a *large* score weight ($\gamma = 15$, top row), the string is driven toward high-likelihood regions, yielding intermediates that lie close to the MEP. Notably, the maximum-likelihood intermediate is a highly abstract, nearly single-color image, confirming that **the MEP passes through likelihood maxima that lie outside the typical set and, as a result, are perceptually unrealistic**.

With a *moderate* score weight ($\gamma = 2$, middle row), intermediates become *cartoon-like*, consistent with prior observa-

Principal curve

Walkers

*Figure 7.* **Principal curve for images from the *goose* class.** Top row: images along the principal curve, computed as the EMA of associated walkers ($T = 0.9$, $\gamma = 7$). Bottom three rows: individual walkers at $t = 1$. The walkers exhibit subtle variations due to entropic effects (best seen when zoomed in); averaging produces the smoother images in the principal curve.

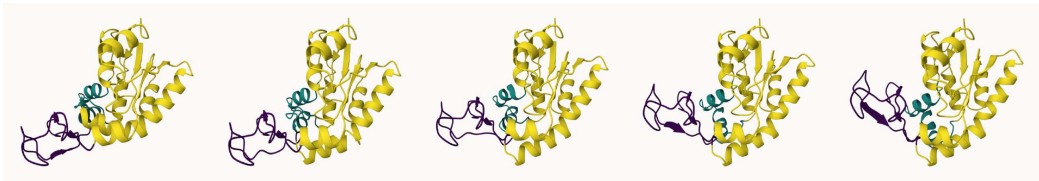

*Figure 8.* **Adenylate Kinase transition pathway.** Pathway between the open (4AKE) and closed (1AKE) conformations computed using DiG. Intermediate structures maintain physical plausibility (secondary structure preservation, no steric clashes).

tions in diffusion-based interpolation (Guth et al., 2025; Karczewski et al., 2025). With *small* score guidance ($\gamma = 10^{-2}$, bottom row), the string produces visually plausible intermediates throughout.

The left panel reports log-likelihood along each string: for large $\gamma$, the trajectory passes through a pronounced likelihood peak. The gap between typical-image likelihoods and this maximum provides a rough estimate of the data-manifold diameter in likelihood space.

**Effect of temperature $T$.** Figure 5 illustrates the finite-temperature method for varying $T$ when $\gamma = 7$.

At *low temperature* ($T = 0.1$, first row), results closely resemble those of the zero-temperature MEP string method, recovering cartoon-like images. At *moderate temperature* ($T = 0.5$, second row), intermediates exhibit slightly more detail but remain cartoon-like. At *high temperature* ($T \approx 1$, final row), **the principal curve passes through realistic-looking intermediates that balance energy and entropy.** Additional examples illustrating the effect of temperature can be found in Appendix H.

**Principal curves.** Figure 7 shows a principal curve between two images from the *goose* class, computed using the

finite-temperature method with $T = 0.9$ and $\gamma = 7$. The first row displays the principal curve itself: each image is obtained by EMA from its associated walker. The remaining three rows show the walkers at $t = 1$ that were used to compute the EMA. Upon close inspection, one can observe subtle differences between walkers—an effect of entropy—which average out to produce the smoother images along the principal curve.

**Likelihood Distributions** In Figure 6, we present the histogram of log-likelihood values for the ImageNet validation set, alongside the log-likelihoods of images obtained from a large collection of strings generated using the algorithms described above. The distribution corresponding to the MEP exhibits pronounced peaks at substantially higher likelihoods than those observed for the validation set. Moreover, the finite-temperature strings display peaks whose locations shift systematically with temperature: as the temperature increases, the peak likelihood decreases, indicating a negative correlation between temperature and likelihood, consistent with theoretical expectations.

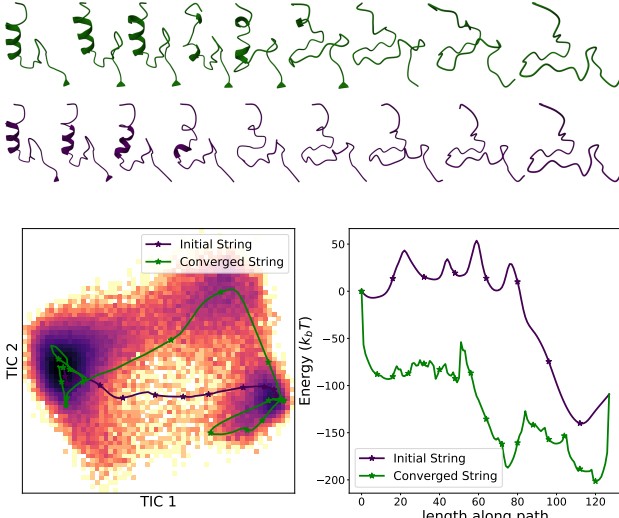

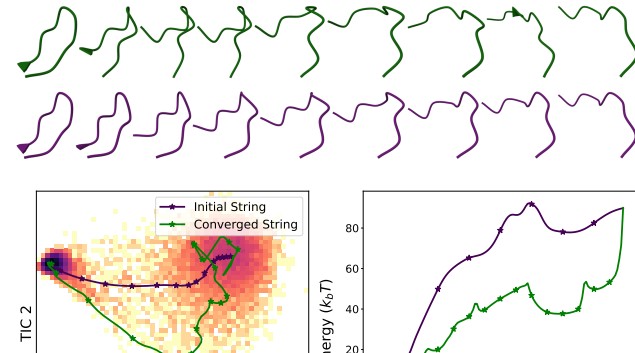

*Figure 9.* **BBA folding pathway.** Top: structures along the initial string obtained by pure transport ($\gamma = 0$). Bottom: structures along the converged MEP. Both show progressive formation of secondary structure. Left panel: initial string (purple) and MEP (green) projected onto the first two TIC components, overlaid on a free-energy landscape estimated from ScoreMD samples; darker regions indicate lower free energy. Asterisks mark equal arc-length intervals. Right panel: energy profile along each pathway. The MEP achieves significantly lower energy than the initial string.

*Figure 10.* **Chignolin folding pathway.** Top: structures along the initial string obtained by pure transport ($\gamma = 0$). Bottom: structures along the converged MEP. Both show progressive formation of secondary structure. Left panel: initial string (purple) and MEP (green) projected onto the first two TIC components, overlaid on a free-energy landscape estimated from ScoreMD samples; darker regions indicate lower free energy. Asterisks mark equal arc-length intervals. Right panel: energy profile along each pathway. The MEP achieves significantly lower energy than the initial string.

## 3.2. Proteins: Conformational Transitions

We apply the string method to predict transition pathways between protein conformations using two diffusion models. One, called DiG (Zheng et al., 2023), operates in SE(3) and it has been trained on experimental structures up to December 2020. The other, ScoreMD (Plainer et al., 2026), operates in $\mathbb{R}^3$ and includes a Fokker–Planck regularization term during training to improve score estimation near $t = 1$.

**Motivation.** Proteins fluctuate between metastable conformations, but experiments typically capture only static snapshots. Transition pathways—the sequence of intermediate structures connecting conformers—are crucial for understanding function but are rarely observed directly. Computational methods such as molecular dynamics can in principle reveal these pathways, but the timescales involved often exceed what is computationally accessible (Shaw et al., 2010). We demonstrate that our framework, applied to a pretrained generative model, can predict plausible transition pathways by computing principal curves with the finite-temperature string method. These pathways connect endpoint conformations while remaining within the typical set of the learned distribution, balancing likelihood and entropy.

**Method.** Given two conformations of the same protein, we apply the string method in the SE(3)-equivariant space of the model. The score function is derived from the model's

denoising objective. Details are given in Appendices F and E.

**Adenylate Kinase.** Adenylate kinase (AdK) is a phosphotransferase enzyme that undergoes a large-scale conformational change between open and closed states during catalysis (Müller et al., 1996). This transition has been extensively studied as a model system for understanding protein dynamics (Arora & Brooks III, 2007; Beckstein et al., 2009). Using DiG, we computed the pathway between the open (PDB: 4AKE) and closed (PDB: 1AKE) conformations (Figure 8). The intermediates preserve secondary structure elements and avoid steric clashes, suggesting physically plausible transitions despite the model being trained only on static structures.

**BBA and Chignolin.** BBA and Chignolin are small protein domains widely used as model systems for protein folding due to their rapid folding kinetics and simple topologies (Lindorff-Larsen et al., 2011). Using ScoreMD (Plainer et al., 2026), we computed minimum energy paths (MEPs) connecting extended and folded conformations for both proteins.

For ScoreMD, the score function is accurately learned near $t = 1$, allowing us to apply the classical string method at a fixed time to compute MEPs. However, the high dimensionality and rough landscape make optimization sensitive to initialization. We therefore use as initialization a string

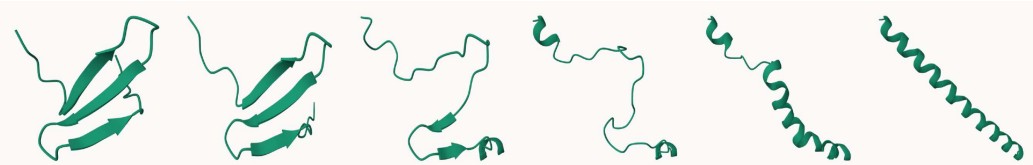

*Figure 11.* Pathway from extended to folded conformation computed using ConfDiff. Intermediates show progressive formation of secondary structure.

obtained by pure transport ($\gamma = 0$).

Figures 9 and 10 show the computed pathways projected onto the first two time-independent components (TICs) (Liu & Chan, 2005). Left panels display the MEP (yellow) and initial string (purple) projected onto TIC coordinates and overlaid on a free-energy landscape, where free energy is estimated as $-\log \rho$ from a histogram of i.i.d. samples generated by ScoreMD; darker regions indicate lower free energy. The MEP is driven toward lower free-energy regions compared to the initial string. Some path segments may appear to overlap due to projection from high dimensions; asterisks mark equal arc-length intervals to clarify the geometry.

Right panels show the free energy profile along each pathway, relative to the starting conformation. The initial string has significantly higher free energy than the converged MEP, demonstrating that the string relaxation is essential. Because the dimensionality is substantially lower than in the image setting, the MEPs remain physically realistic rather than cartoonish.

Above the two panels for each protein, we include a three-dimensional rendering of its folding pathway; the protein is colored consistently with the corresponding pathway to facilitate visual correspondence.

**WW Domain.** The WW domain is a small protein module that has served as a model system for studying protein folding due to its fast folding kinetics and simple topology (Jäger et al., 2006). Using ConfDiff (Wang et al., 2024), we computed the folding pathway from an extended to a folded conformation (Figure 11). The pathway reveals progressive formation of secondary structure, consistent with experimental observations of WW domain folding (Liu & Chan, 2005).

Enlarged figures and experimental details are presented in Appendix G.

## 4. Conclusion

We adapted the string method to probe the geometry of learned distributions using score functions from pretrained generative models. By varying the dynamics—pure transport, gradient-dominated, or finite-temperature—we reveal complementary aspects of the distribution landscape.

Our key finding is that accounting for entropy is essential in high dimensions. Minimum energy paths traverse high-likelihood but low-probability regions, producing cartoonish artifacts. Principal curves, which balance energy against entropy, yield realistic transitions with stronger theoretical grounding. These results confirm and explain prior observations that high-likelihood samples from diffusion models appear unrealistic (Guth et al., 2025; Karczewski et al., 2025): the phenomenon is not a model defect but a consequence of concentration of measure, and can be resolved by accounting for entropy.

This establishes the string method as a tool for analyzing generative models beyond sampling: identifying modes, characterizing barriers, and mapping connectivity in complex learned distributions. Importantly, the method requires no retraining—only access to the learned velocity and score fields.

**Limitations.** The computed pathways are only as good as the underlying generative model. If the model assigns low probability to physically relevant transition states, the string method cannot recover them. Similarly, errors in score estimation—particularly near the data distribution—propagate to the computed paths, motivating our quenching strategy.

**Future directions.** Several extensions are natural. Scaling to larger models (e.g., text-to-image diffusion) would test whether the likelihood-realism tradeoff persists across architectures. Theoretical analysis of convergence rates and approximation error would strengthen the foundations. Beyond images and proteins, the method applies wherever transition pathways matter: molecular design, robotics, and latent space exploration in multimodal models. Finally, combining string methods with conditional generation could enable targeted pathway computation—for instance, finding transitions that pass through specified intermediate states.

## Impact Statement

This paper presents work whose goal is to advance the field of Machine Learning. There are many potential societal consequences of our work, none which we feel must be specifically highlighted here.

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

# A. Background on Diffusion Models

## A.1. Velocity and Score: Definitions and Estimators

In the stochastic interpolant framework, we construct a time-dependent density $\rho_t$ via the interpolant $I_t = \alpha_t x_0 + \beta_t x_1$ with $x_0 \sim \mathcal{N}(0, I)$ and $x_1 \sim \rho_1$. Common choices include:

- Linear: $\alpha_t = 1 - t$, $\beta_t = t$

- Trigonometric: $\alpha_t = \cos(\frac{\pi t}{2})$, $\beta_t = \sin(\frac{\pi t}{2})$ (variance-preserving)

- $\alpha_t = \sqrt{1 - t^2}$, $\beta_t = t$ (variance-preserving; time-rescaled Ornstein-Uhlenbeck)

The velocity field and score are defined as:

$$b_t(x) = \mathbb{E}[\dot{I}_t \mid I_t = x], \tag{8}$$
$$s_t(x) = \nabla \log \rho_t(x). \tag{9}$$

Both can be estimated by regression: $b_t$ by minimizing $\mathbb{E}[|\dot{I}_t - b_t(I_t)|^2]$, and $s_t$ via denoising score matching using the conditional Gaussian structure $I_t \mid x_1 \sim \mathcal{N}(\beta_t x_1, \alpha_t^2 I)$. The two are related by:

$$s_t(x) = \frac{\beta_t b_t(x) - \dot{\beta}_t x}{\alpha_t(\dot{\alpha}_t \beta_t - \alpha_t \dot{\beta}_t)}. \tag{10}$$

## A.2. The General SDE and Fokker-Planck Equation

The general SDE used in our framework is:

$$dx_t = b_t(x_t)\, dt + \gamma_t^2 s_t(x_t)\, dt + \sqrt{2}\gamma_t\, dW_t. \tag{11}$$

The corresponding Fokker-Planck equation for the density is:

$$\partial_t \rho_t + \nabla \cdot \left[(b_t + \gamma_t^2 s_t)\rho_t\right] = \gamma_t^2 \Delta \rho_t. \tag{12}$$

Substituting $s_t = \nabla \log \rho_t$, one can verify that $\rho_t$ is preserved for any $\gamma_t \geq 0$: the score correction $\gamma_t^2 s_t \rho_t = \gamma_t^2 \nabla \rho_t$ and the diffusion term $\gamma_t^2 \Delta \rho_t$ cancel in the flux, leaving the evolution determined by $b_t$ alone. This is a local detailed balance condition at each time $t$; since $b_t$ and $s_t$ are time-dependent, the overall dynamics describe nonequilibrium sampling from $\rho_0$ to $\rho_1$.

## A.3. Likelihood Computation via the Probability Flow ODE

Setting $\gamma_t = 0$ in the general SDE gives the probability flow ODE

$$\dot{x}_t = b_t(x_t). \tag{13}$$

The corresponding transport equation for the density is

$$\partial_t \rho_t + \nabla \cdot (b_t \rho_t) = 0. \tag{14}$$

This can be solved by the method of characteristics. Let $x_t$ be a solution of the ODE starting from $x_0$. Differentiating $\rho_t(x_t)$ along the flow we deduce

$$\frac{d}{dt}\rho_t(x_t) = \partial_t \rho_t(x_t) + \nabla \rho_t(x_t) \cdot \dot{x}_t = \partial_t \rho_t(x_t) + \nabla \rho_t(x_t) \cdot b_t(x_t). \tag{15}$$

Using the transport equation $\partial_t \rho_t = -\nabla \cdot (b_t \rho_t) = -b_t \cdot \nabla \rho_t - \rho_t \nabla \cdot b_t$, we obtain

$$\frac{d}{dt}\rho_t(x_t) = -\rho_t(x_t)\nabla \cdot b_t(x_t). \tag{16}$$

This is a linear ODE in $\rho_t(x_t)$, with solution

$$\rho_t(x_t) = \rho_0(x_0) \exp\left(-\int_0^t \nabla \cdot b_\tau(x_\tau)\, d\tau\right). \tag{17}$$

Taking logarithms gives the log-likelihood

$$\log \rho_t(x_t) = \log \rho_0(x_0) - \int_0^t \nabla \cdot b_\tau(x_\tau)\, d\tau. \tag{18}$$

In practice, given a data sample $x_1 \sim \rho_1$, we integrate the ODE backward from $t = 1$ to $t = 0$ to obtain $x_0$, and accumulate the divergence $\nabla \cdot b_t$ along the trajectory. Since $\rho_0 = \mathcal{N}(0, I)$, the term $\log \rho_0(x_0)$ is simply $-\frac{1}{2}\|x_0\|^2 - \frac{d}{2}\log(2\pi)$.

## B. String Method: Derivation and Implementation

### B.1. Continuous Evolution

We derive the continuous string evolution equation from the constraint that arc-length parametrization is preserved.

**Proposition B.1.** *Let $\phi_t : [0, 1] \to \mathbb{R}^d$ be a curve evolving under a velocity field $v_t$. Assume that the endpoints evolve as independent points:*

$$\dot{\phi}_t(0) = v_t(\phi_t(0)), \quad \dot{\phi}_t(1) = v_t(\phi_t(1)). \tag{19}$$

*Then, for $s \in (0, 1)$, the evolution*

$$\dot{\phi}_t(s) = v_t(\phi_t(s)) + \lambda_t(s)\partial_s\phi_t(s) \tag{20}$$

*preserves the arc-length parametrization $|\partial_s\phi_t(s)| = L(t)$ (constant in $s$) for an appropriate choice of $\lambda_t(s)$ with $\lambda_t(0) = \lambda_t(1) = 0$.*

*Proof.* The arc-length parametrization requires $|\partial_s\phi_t|^2$ to be constant in $s$, i.e., $\langle \partial_s\phi_t, \partial_s^2\phi_t \rangle = 0$ for all $s$ and $t$. Taking the time derivative:

$$\partial_t \langle \partial_s\phi_t, \partial_s^2\phi_t \rangle = \langle \partial_s\dot{\phi}_t, \partial_s^2\phi_t \rangle + \langle \partial_s\phi_t, \partial_s^2\dot{\phi}_t \rangle = 0. \tag{21}$$

Substituting $\dot{\phi}_t = v_t(\phi_t) + \lambda_t\partial_s\phi_t$ and requiring this to hold for all $s$ determines $\lambda_t(s)$. The boundary conditions $\lambda_t(0) = \lambda_t(1) = 0$ are consistent with the endpoints evolving according to $v_t$. $\qquad\square$

In practice, we enforce the constraint via discrete reparametrization rather than computing $\lambda_t$ explicitly. The derivation above applies to $\mathbb{R}^d$ with the Euclidean metric; for Riemannian manifolds such as $SO(3)$, we work directly with the discrete algorithm.

### B.2. Reparametrization on $\mathbb{R}^d$

The discrete reparametrization step redistributes points $\{\phi^{(i)}\}_{i=0}^N$ to equal arc-length spacing:

1. Compute cumulative arc-lengths: $L_0 = 0$, $L_i = L_{i-1} + |\phi^{(i)} - \phi^{(i-1)}|$.

2. Normalize: $\alpha_i = L_i/L_N \in [0, 1]$.

3. Fit a cubic spline through $(\alpha_i, \phi^{(i)})$.

4. Evaluate at uniform positions: $\phi_{\text{new}}^{(i)} = \text{spline}(i/N)$.

### B.3. Reparametrization on $SO(3)$

On $SO(3)$, reparametrization requires converting between the rotation matrix representation $\{\phi_M^{(i)}\}_{i=0}^N$ and the axis-angle vector representation $\{\phi_V^{(i)}\}_{i=0}^N$. To reparametrize into $K + 1$ equally spaced points $\{\psi^{(j)}\}_{j=0}^K$:

1. Compute the incremental rotations: $s_M^{(i)} = \phi_M^{(i)}(\phi_M^{(i-1)})^{-1}$ for $1 \leq i \leq N$.

2. Compute cumulative arc-lengths in the axis-angle representation: $L_0 = 0$, $L_i = L_{i-1} + |s_V^{(i)}|$.

3. Normalize: $\alpha_i = L_i/L_N \in [0,1]$.

4. For each $0 \le j \le K$, find the preceding index $p(j) = \max\{i : \alpha_i \le j/K\}$.

5. Compute interpolated displacements: $d_V^{(j)} = s_V^{(p(j)+1)} \cdot \frac{j/K - \alpha_{p(j)}}{\alpha_{p(j)+1} - \alpha_{p(j)}}$.

6. Set $\psi_M^{(0)} = \phi_M^{(0)}$, $\psi_M^{(K)} = \phi_M^{(N)}$, and for $1 \le j \le K-1$: $\psi_M^{(j)} = d_M^{(j)} \phi_M^{(p(j))}$.

### B.4. Reparametrization on $SE(3)$

The group $SE(3)$ is the semidirect product of $\mathbb{R}^3$ and $SO(3)$, hence every element $\phi \in SE(3)$ can be written as $\phi = (t, R)$ with $t \in \mathbb{R}^3$ and $R \in SO(3)$. There is not a criterion to choose a norm in $SE(3)$, in this paper we chose to have $|\phi| = \sqrt{|t|^2 + |R_V|^2}$ where $|R_V|$ is the norm of the vector in the axis angle representation. With this choice for the norm we do the reparameterization as follows:

1. Compute the incremental rotations: $s_M^{(i)} = R_M^{(i)}(R_M^{(i-1)})^{-1}$ for $1 \le i \le N$.

2. Compute the incremental translations: $q^{(i)} = t^{(i)} - t^{(i-1)}$ for $1 \le i \le N$.

3. Compute cumulative arc-lengths in with this norm: $L_0 = 0$, $L_i = L_{i-1} + \sqrt{|q|^2 + |s_V|^2}$.

4. Normalize: $\alpha_i = L_i/L_N \in [0,1]$.

5. For each $0 \le j \le K$, find the preceding index $p(j) = \max\{i : \alpha_i \le j/K\}$.

6. Compute interpolated rotation displacements: $d_V^{(j)} = s_V^{(p(j)+1)} \cdot \frac{j/K - \alpha_{p(j)}}{\alpha_{p(j)+1} - \alpha_{p(j)}}$.

7. Compute interpolated translation displacements: $y^{(j)} = q^{(p(j)+1)} \cdot \frac{j/K - \alpha_{p(j)}}{\alpha_{p(j)+1} - \alpha_{p(j)}}$.

8. Set $\psi_M^{(0)} = \phi_M^{(0)}$, $\psi_M^{(K)} = \phi_M^{(N)}$, and for $1 \le j \le K-1$: $\psi_M^{(j)} = (t^{p(j)} + y^{(j)}, d_M^{(j)} \phi_M^{(p(j))})$.

## C. Testing the score reliability on Gaussian Mixtures

To examine how score estimation error varies along the generative process (i.e., as $t$ increases from 0 to 1), we trained MLPs to learn a stochastic interpolant transporting a Gaussian distribution to a mixture of two Gaussians in $d$ dimensions, for different values of $d$. The two modes in the mixture have means $(\pm 3.0, 0, \ldots, 0) \in \mathbb{R}^d$ and covariance matrices

$$\begin{pmatrix} \frac{7.9}{4} & \pm\frac{6.7\sqrt{3}}{4} \\ \pm\frac{6.7\sqrt{3}}{4} & \frac{7.9}{4} \end{pmatrix} \oplus I_{d-2}.$$

where the sign corresponds to the sign of the mean.

We parametrized the score function using an MLP with 3 hidden layers of sizes 512, 1024, and 512. Each model was trained with batch size 1000 for $1500 \times d$ iterations. For inference, we used batch size $5000 \times d$ and 200 timesteps. For each value of $d$, we trained 10 models; Figure 3 shows the average relative score error $\mathbb{E}[|s_t - \hat{s}_t|/|s_t|]$ as a function of $t$.

## D. Hyperparameters for SiT

*Table 2.* Hyperparameter choices for the SiT Model

| Hyperparameters | Values |
|---|---|
| **Neural network** | |
| Depth | 28 |
| Hidden Size | 1152 |
| Patch Size | 2 |
| Number of Attention Heads | 16 |
| MLP ratio | 4.0 |
| Class Dropout Probability | 0.1 |
| Input Size | 32 |
| Input Channels | 4 |

## E. Hyperparameters for DiG

*Table 3.* Hyperparameters of the Distributional Graphormer Protein Model.

| Hyperparameter | Initialization | PIDP | Data Training |
|---|---|---|---|
| Model depth | | 12 | |
| Hidden dim (Single) | | 768 | |
| Hidden dim (Pair) | | 256 | |
| Hidden dim (Feed Forward) | | 1024 | |
| Number of Heads | | 32 | |

## F. Hyperparameters for ScoreMD

*Table 4.* **Graph transformer model definitions** extracted from the configuration file.

| Model name | hidden_nf | feature_embedding_dim | n_layers | potential | dropout |
|---|---|---|---|---|---|
| transformer_large_score | 128 | 16 | 3 | false | 0.0 |
| transformer_large_potential | 128 | 16 | 3 | true | 0.0 |

*Table 5.* **Ranged models configuration** extracted from the configuration file of the model, it should be kept in mind that for this model $t = 0$ is actually the interpolant time $t = 1$ and vice versa.

| Entry | Model reference | Range |
|---|---|---|
| 1 | `transformer_large_score` | $[1.0,\ 0.6]$ |
| 2 | `transformer_large_score` | $[0.6,\ 0.1]$ |
| 3 | `transformer_large_potential` | $[0.1,\ 0.0]$ |

# G. Details of the Experiments Proteins

## G.1. Hyperparameters of the model

*Table 6.* Hyperparameter choices for ConfDiff Model

| Hyperparameters | Values |
|---|---|
| **Neural network** | |
| Number of IPA blocks | 4 |
| Dimension of single repr. | 256 |
| Dimension of pairwise Repr. | 128 |
| Dimension of hidden | 256 |
| Number of IPA attention heads | 4 |
| Number of IPA query points | 8 |
| Number of IPA value points | 12 |
| Number of transformer attention heads | 4 |
| Number of transformer layers | 2 |

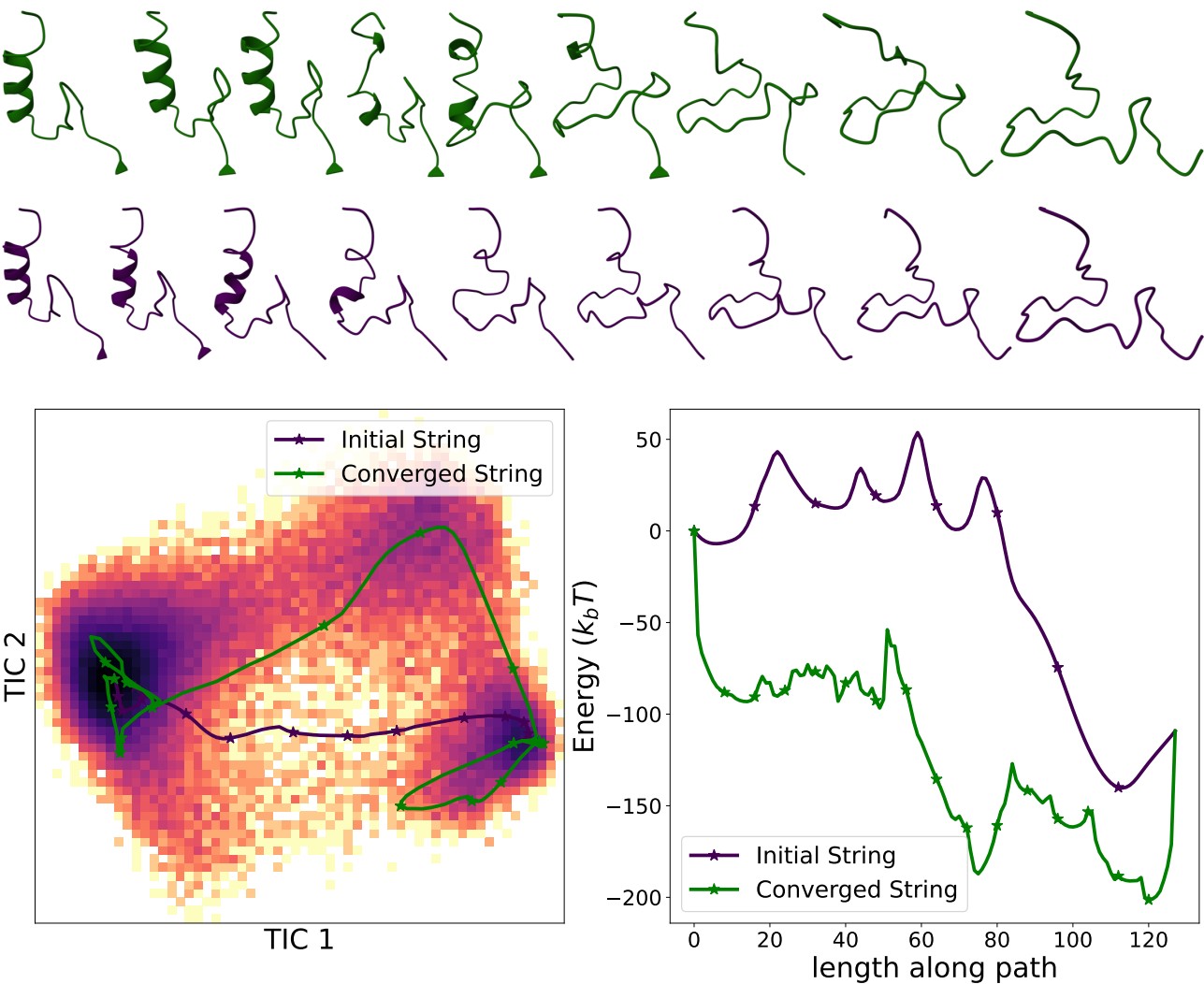

*Figure 12.* **BBA folding pathway.** Top: structures along the initial string obtained by pure transport ($\gamma = 0$). Bottom: structures along the converged MEP. Both show progressive formation of secondary structure. Left panel: initial string (purple) and MEP (green) projected onto the first two TIC components, overlaid on a free-energy landscape estimated from ScoreMD samples; darker regions indicate lower free energy. Asterisks mark equal arc-length intervals. Right panel: energy profile along each pathway. The MEP achieves significantly lower energy than the initial string.

# H. Additional Experiments

## H.1. Effect of the Temperature in finite-temperature string method

## H.2. Multiple realizations of a principle curves

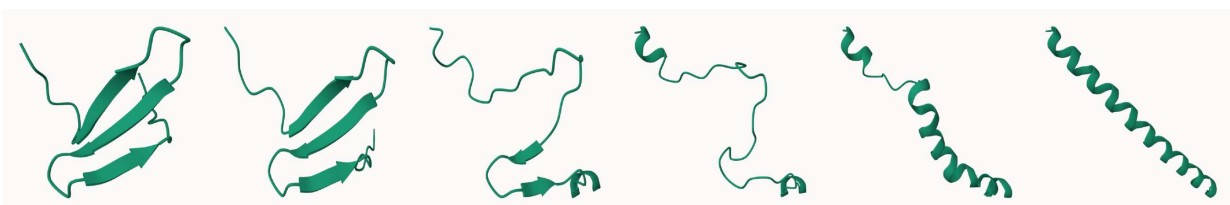

*Figure 13.* **Chignolin folding pathway.** Top: structures along the initial string obtained by pure transport ($\gamma = 0$). Bottom: structures along the converged MEP. Both show progressive formation of secondary structure. Left panel: initial string (purple) and MEP (green) projected onto the first two TIC components, overlaid on a free-energy landscape estimated from ScoreMD samples; darker regions indicate lower free energy. Asterisks mark equal arc-length intervals. Right panel: energy profile along each pathway. The MEP achieves significantly lower energy than the initial string.

*Figure 14.* Pathway from extended to folded conformation computed using ConfDiff. Intermediates show progressive formation of secondary structure.

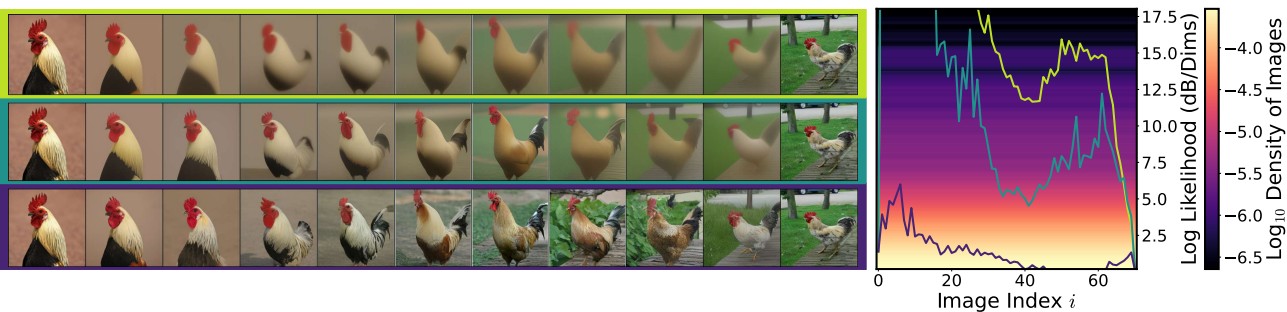

*Figure 15.* Effect of temperature $T$ on images principal curve. Lower $T$ drives paths through cartoonish high-likelihood modes (top); higher $T$ preserves realism (bottom).

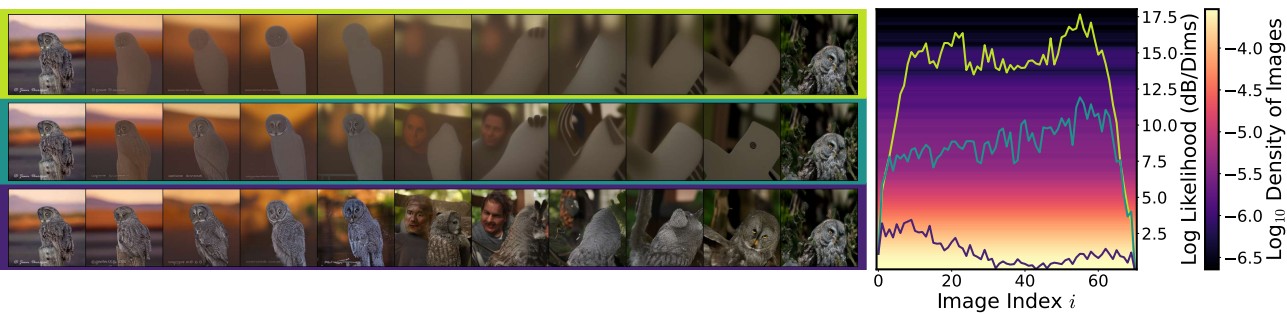

*Figure 16.* Effect of temperature $T$ on images principal curve. Lower $T$ drives paths through cartoonish high-likelihood modes (top); higher $T$ preserves realism (bottom).

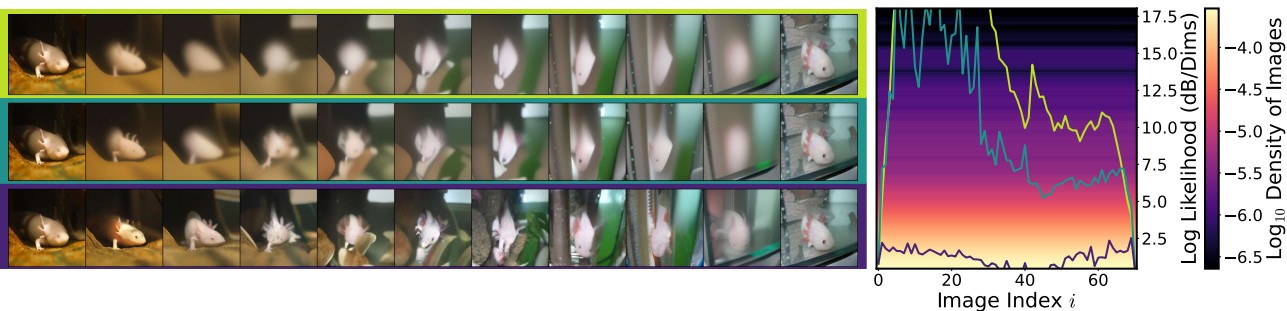

*Figure 17.* Effect of temperature $T$ on images principal curve. Lower $T$ drives paths through cartoonish high-likelihood modes (top); higher $T$ preserves realism (bottom).

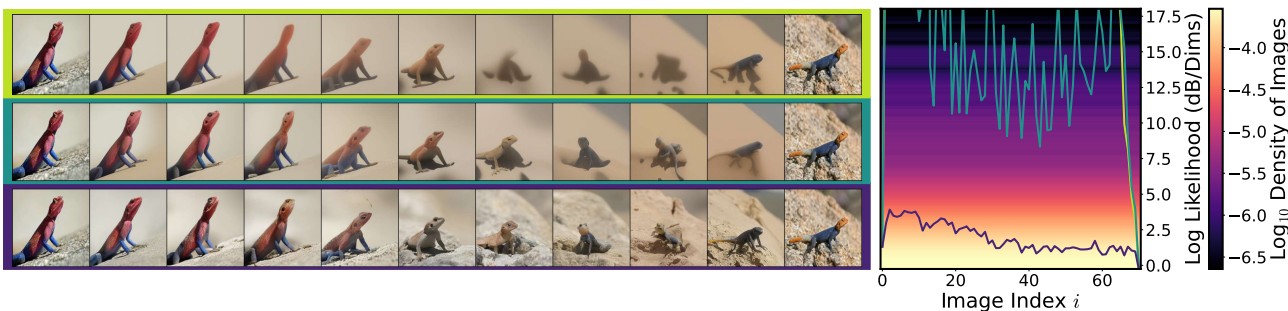

*Figure 18.* Effect of temperature $T$ on images principal curve. Lower $T$ drives paths through cartoonish high-likelihood modes (top); higher $T$ preserves realism (bottom).

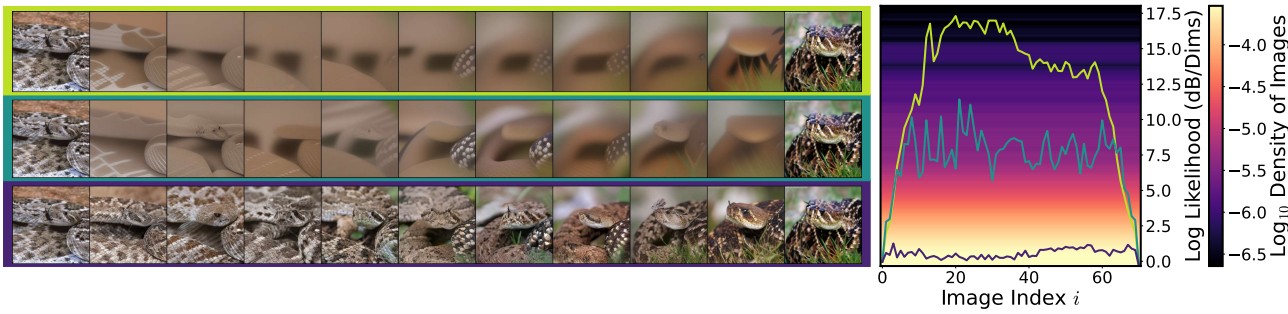

*Figure 19.* Effect of temperature $T$ on images principal curve. Lower $T$ drives paths through cartoonish high-likelihood modes (top); higher $T$ preserves realism (bottom).

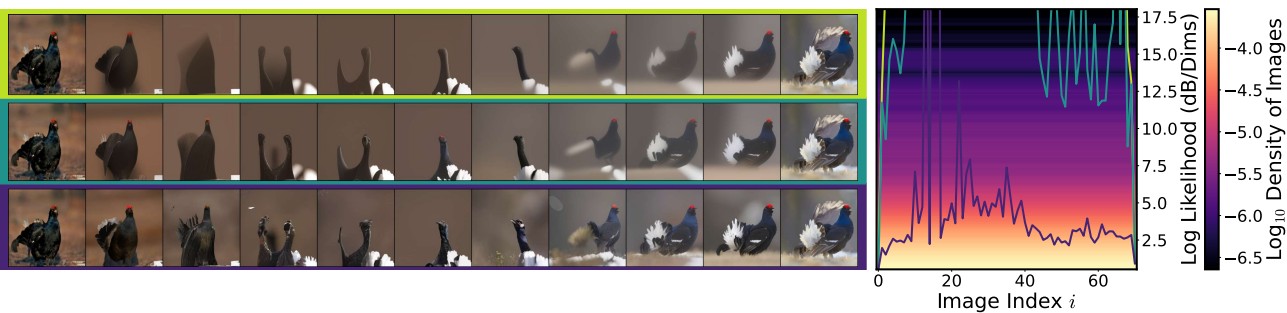

*Figure 20.* Effect of temperature $T$ on images principal curve. Lower $T$ drives paths through cartoonish high-likelihood modes (top); higher $T$ preserves realism (bottom).

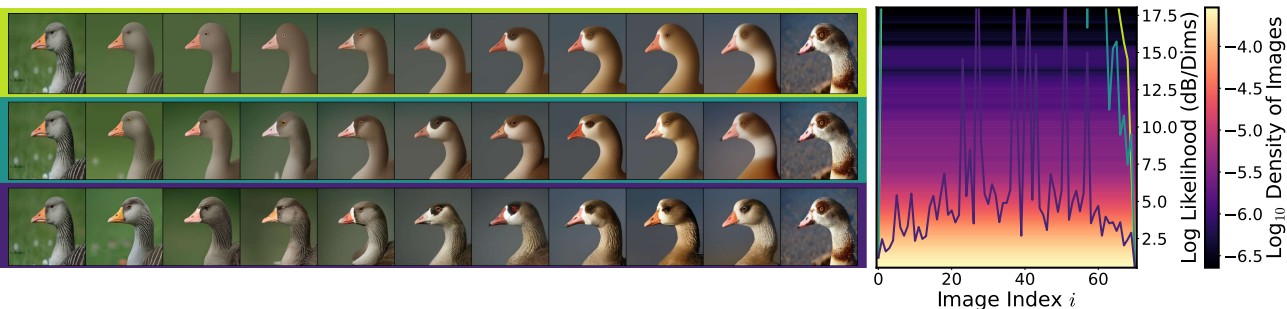

*Figure 21.* Effect of temperature $T$ on images principal curve. Lower $T$ drives paths through cartoonish high-likelihood modes (top); higher $T$ preserves realism (bottom).

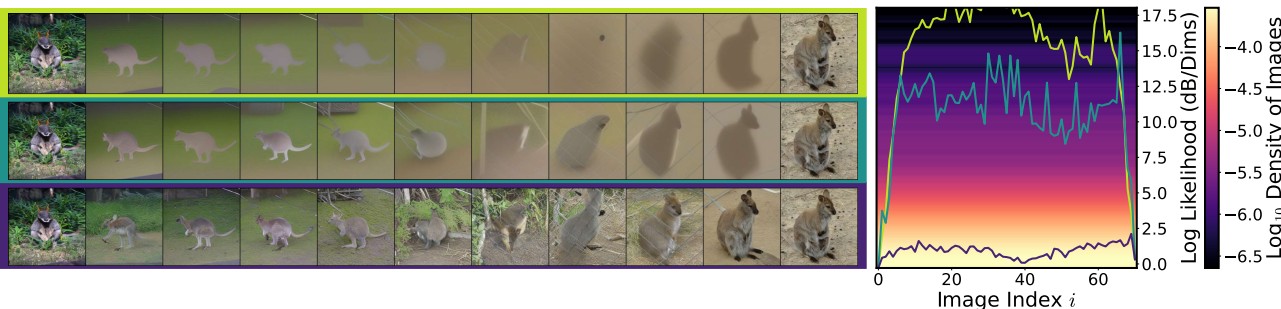

*Figure 22.* Effect of temperature $T$ on images principal curve. Lower $T$ drives paths through cartoonish high-likelihood modes (top); higher $T$ preserves realism (bottom).

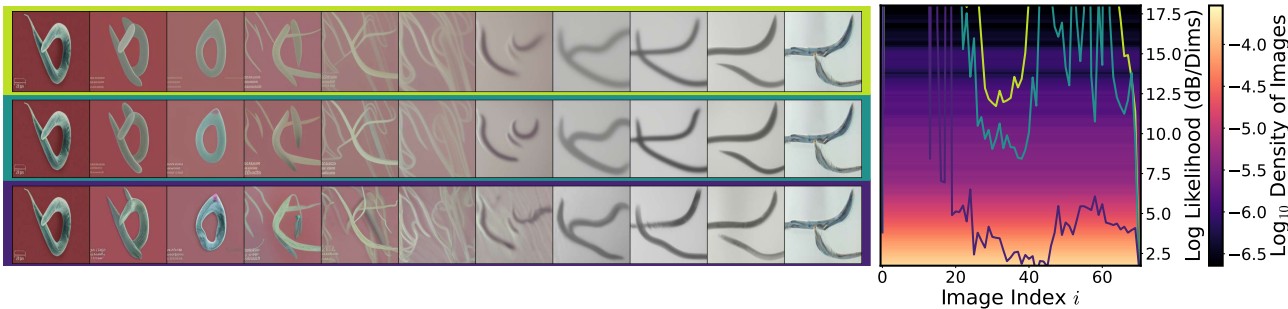

*Figure 23.* Effect of temperature $T$ on images principal curve. Lower $T$ drives paths through cartoonish high-likelihood modes (top); higher $T$ preserves realism (bottom).

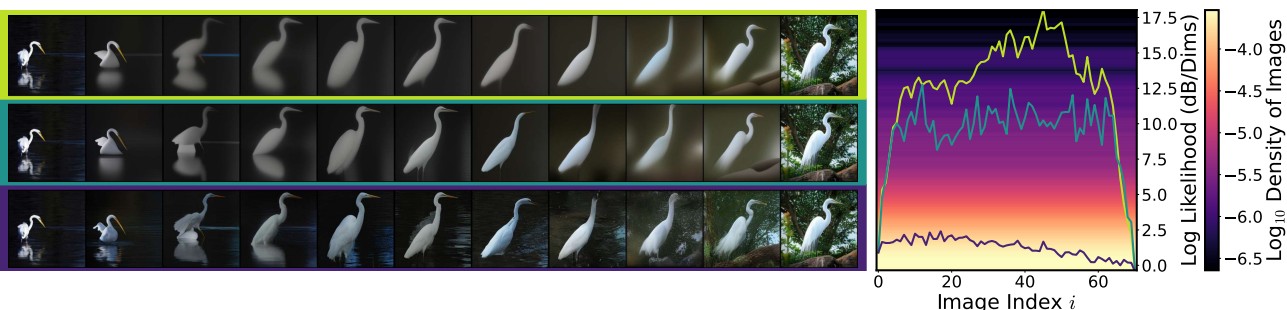

*Figure 24.* Effect of temperature $T$ on images principal curve. Lower $T$ drives paths through cartoonish high-likelihood modes (top); higher $T$ preserves realism (bottom).

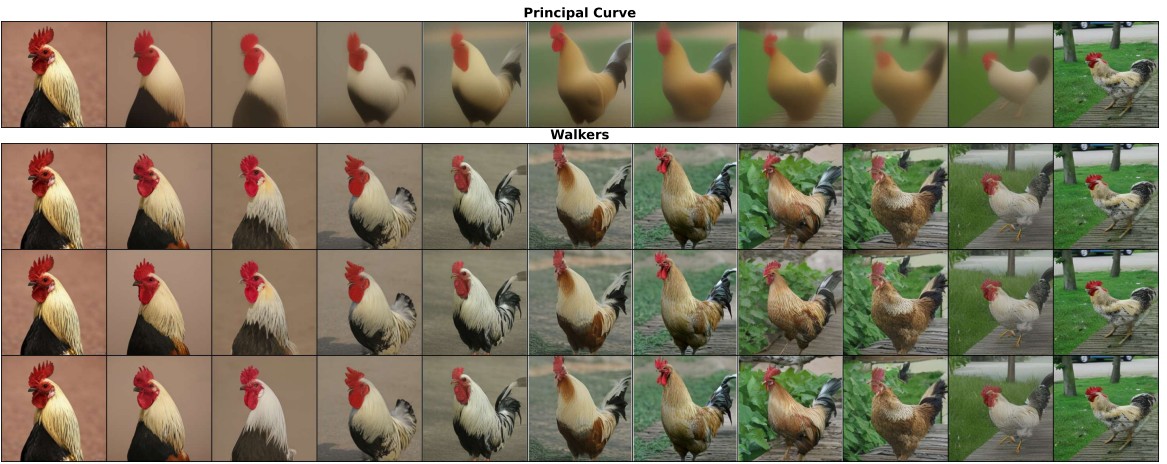

*Figure 25.* A principal curves with three realizations coming from this curve

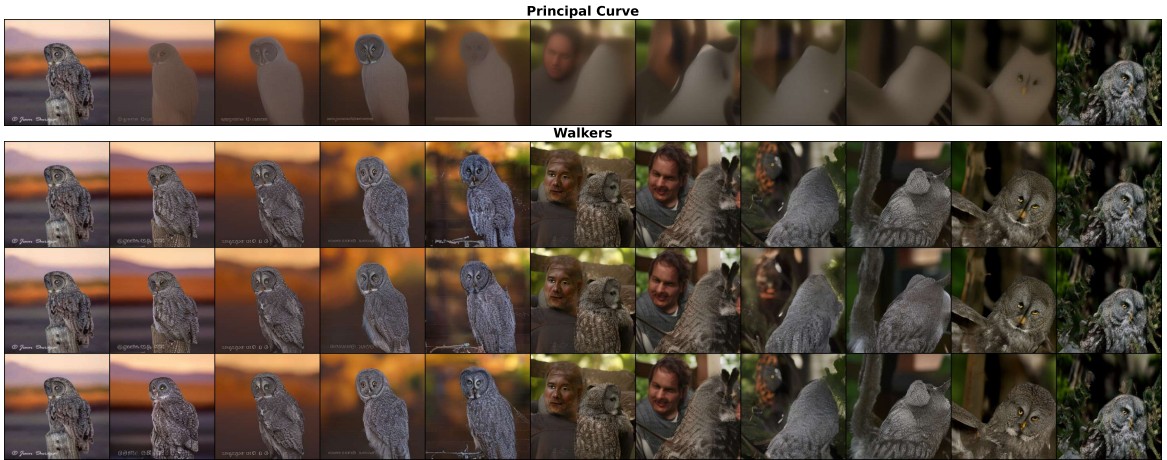

*Figure 26.* A principal curves with three realizations coming from this curve

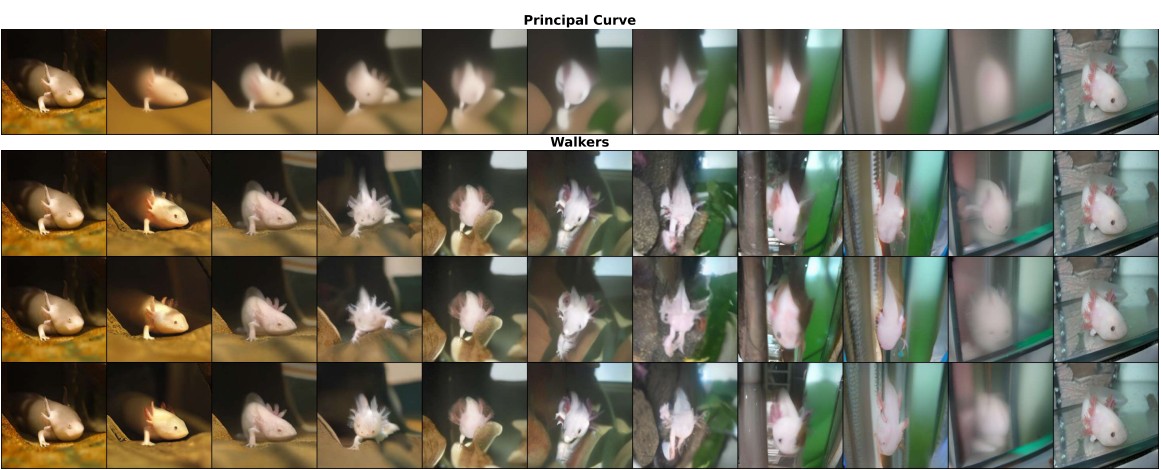

*Figure 27.* A principal curves with three realizations coming from this curve

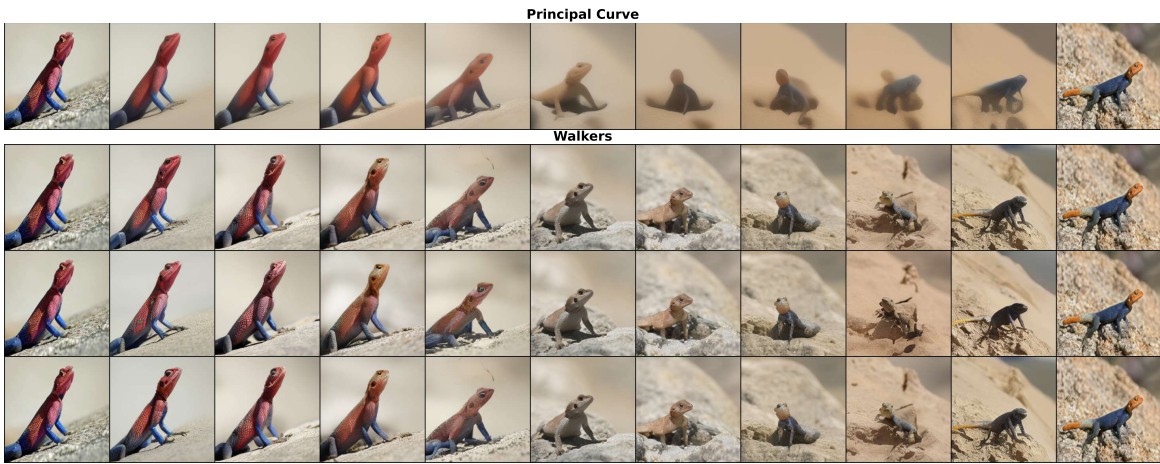

*Figure 28.* A principal curves with three realizations coming from this curve

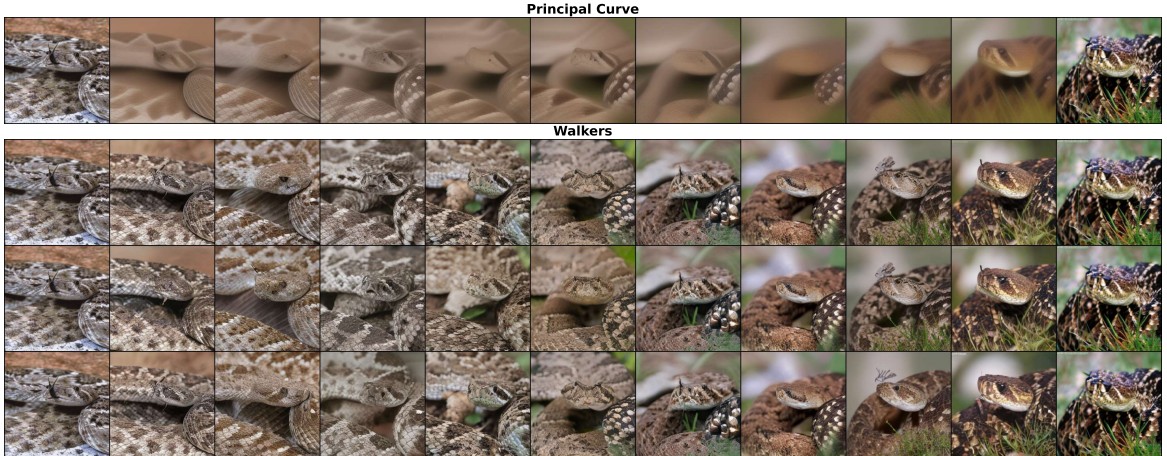

*Figure 29.* A principal curves with three realizations coming from this curve

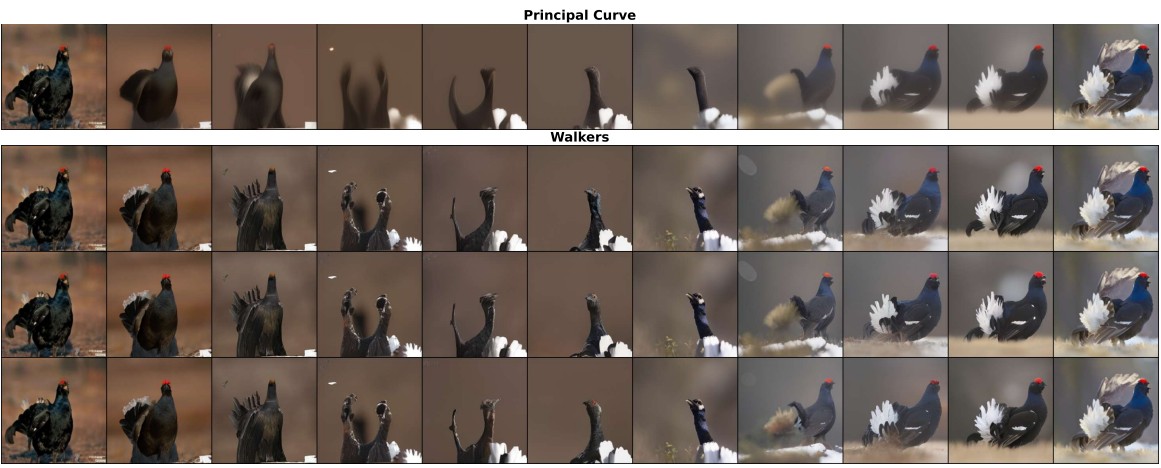

*Figure 30.* A principal curves with three realizations coming from this curve

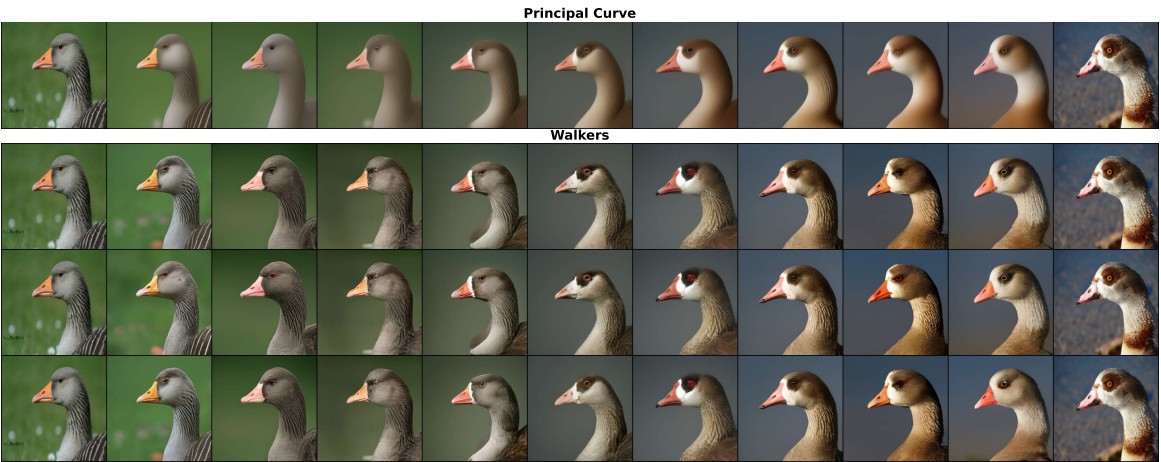

*Figure 31.* A principal curves with three realizations coming from this curve

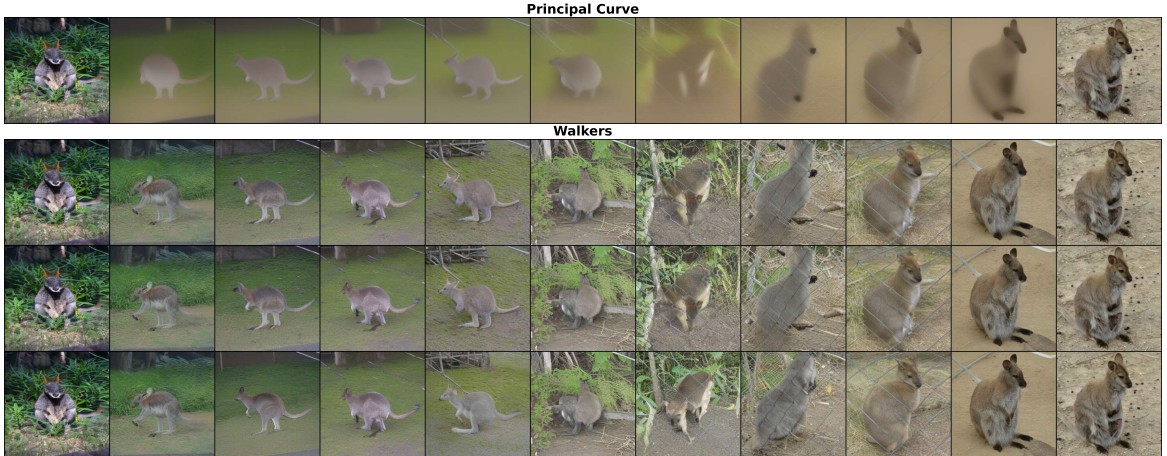

*Figure 32.* A principal curves with three realizations coming from this curve

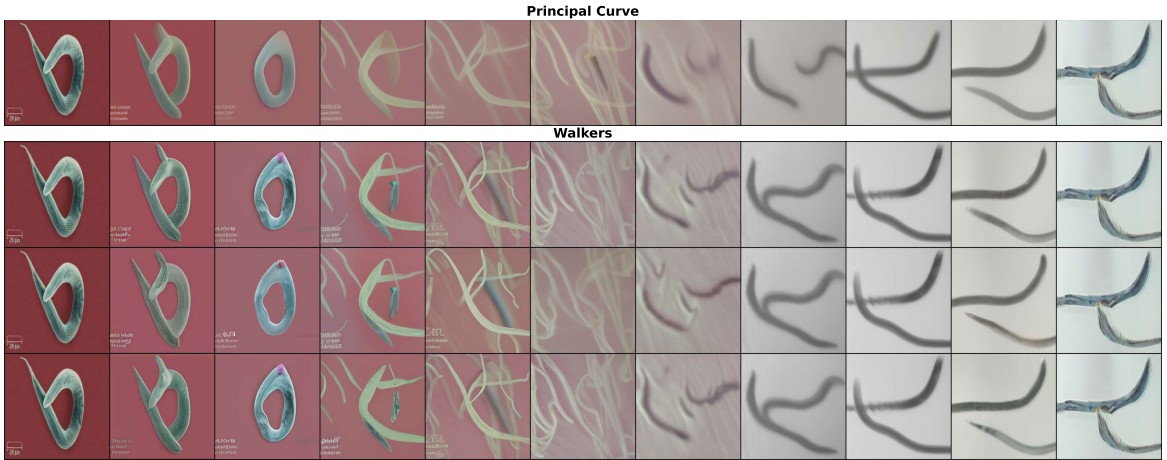

*Figure 33.* A principal curves with three realizations coming from this curve

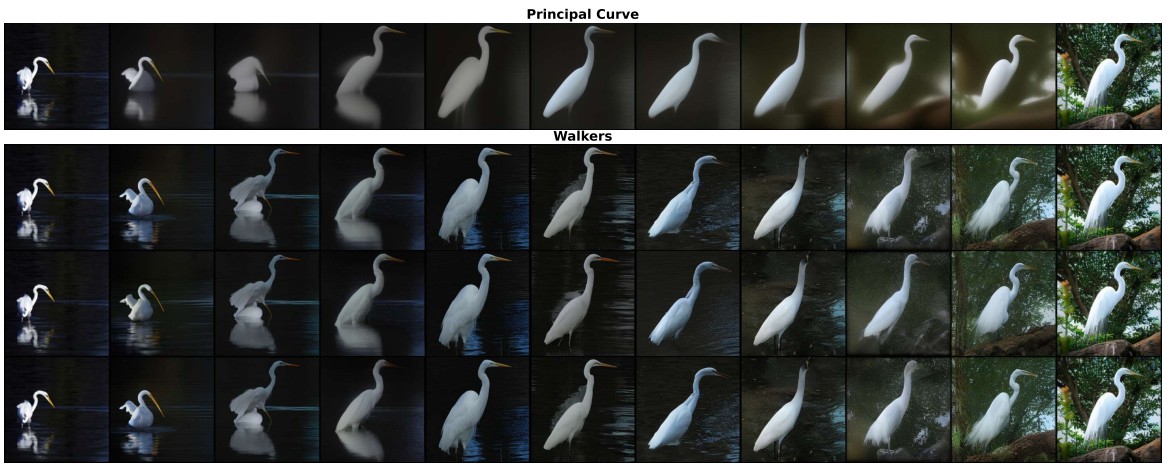

*Figure 34.* A principal curves with three realizations coming from this curve

