# OpenReview forum: "Probing the Geometry of Diffusion Models with the String Method"
_ICML.cc/2026/Conference — ICML 2026 regular_

### Official Review · Reviewer_RRHn · 2026-03-11

**Soundness:** 3
**Presentation:** 4
**Significance:** 3
**Originality:** 3
**Overall Recommendation:** 5
**Confidence:** 3

**Summary:**

This paper proposes a novel approach to interpolation within pretrained generative models by adapting the "string method",  a technique originally developed in computational chemistry for probing transition paths. The authors establish a solid mathematical foundation to compare three distinct interpolation regimes: pure transport, Minimum Energy Paths (MEP), and principal curves. Through experiments across different modalities (images and molecular conformations), the work demonstrates the advantages of the finite-temperature string method. Additionally, the authors provide an intuitive and valuable theoretical interpretation of the "likelihood-realistic paradox" and the "cartoonish phenomenon" from the perspective of measure concentration.

**Compliance With Llm Reviewing Policy:**

Affirmed.

**Final Justification:**

The authors use the string method from computational chemistry to explore the energy landscape of generative models and construct an interpolation path between two images. The authors' rebuttal has addressed all my concerns, so I am raising my score to 5.

**Key Questions For Authors:**

- 1.Clarification on "Max-Likelihood" (Line 206). Could the authors clarify the precise definition of the "max-likelihood" of a path? Does this refer to the integral of the pointwise likelihood along the trajectory, or is it defined differently in this context?
- 2.The discussion on the likelihood-realistic paradox is insightful. However, earlier literature, such as Nalisnick et al. (2019, *"Do Deep Generative Models Know What They Don't Know?"*), has also explored this paradox, particularly in flow-based models. Could the authors briefly elaborate on how their perspective differs from  the arguments presented in that work?
- 3.To strengthen the claims regarding the molecular conformation transition task, could the authors provide standard quantitative metrics to evaluate the plausibility of the generated intermediate states? Comparing the results against relevant baselines would make this section significantly more convincing than relying solely on visual inspection.
- 4.Reparameterizations on Manifolds (Appendix B). Could the authors provide a brief demonstration on why the reparameterizations are reasonable, especially for SO(3) and SE(3)? These seem not trivial.

**Limitations:**

yes

**Strengths And Weaknesses:**

### Strengths
1. Originality and Cross-disciplinary Innovation: The application of the string method from computational chemistry to generative modeling is innovative, providing a fresh and theoretically grounded perspective on continuous interpolation tasks between different modes.
2. Theoretical Grounding: The paper features solid mathematical foundations and well-designed algorithmic implementations.
3. Clarity: The manuscript is exceptionally well-written, logically organized, and free of unnecessary tautologies.

### Weaknesses
Empirical Validation in Molecular Conformation: While the visual results for molecular conformation transformations are interesting, the empirical evaluation in this section relies heavily on qualitative intuition. The absence of quantitative metrics and comparisons with standard baselines makes it difficult to fully assess the method's practical effectiveness and physical plausibility in this specific domain.

---

> ### Author Rebuttal · Authors · 2026-03-30
>
> We thank the reviewer for the positive assessment and for the specific technical questions, which we address in turn.
>
> ---
>
> **Q1 — Definition of "maximum-likelihood path"**
>
> A maximum-likelihood path (MEP) is a curve $\varphi^*: [0,1] \to \mathbb{R}^d$ satisfying:
>
> $$[\nabla V(\varphi^*(s))]^\perp = 0 \quad \text{for all } s \in [0,1],$$
>
> where $[\cdot]^\perp$ denotes the component perpendicular to the tangent $\partial_s \varphi^*$ and $V = -\log \rho_1$ is the data log-density (Definition 1). This means the score has no component pushing the path sideways. Equivalently, the path passes through local maxima of $\rho_1$ restricted to any hyperplane orthogonal to it: the likelihood is locally maximised *transversally* at every point, justifying the name. This definition is standard in computational chemistry (E, Ren & Vanden-Eijnden, 2002) and is distinct from maximising $\int \log \rho_1(\varphi(s))\, ds$. We will add this clarification in the caption of Fig. 2 and in Section 2.3.2.
>
> **Q2 — Relationship to Nalisnick et al. (2019)**
>
> Nalisnick et al. (2019) showed that deep generative models assign higher likelihood to out-of-distribution inputs than to in-distribution ones — a manifestation of the same concentration-of-measure phenomenon we exploit: in high dimensions, the typical set is far from the mode. Their observation was framed as a failure mode of likelihood-based anomaly detection.
>
> Our perspective differs in two ways: we treat this as a structural property to be exploited rather than a failure mode, using it to explain why MEPs produce cartoonish images and to motivate the principal-curve regime as a remedy; and we make the connection explicit geometrically — the gap is directly observable as the likelihood peak traversed by MEP paths (Figs. 3–4), disappearing when entropy is accounted for via temperature.
>
> **Q3 — Quantitative metrics for protein conformational transitions**
>
> We have conducted new experiments available in the anonymous repository: https://anonymous.4open.science/r/String-Rebuttal-7E21/rebuttal_images_organized.pdf.
>
> - **TICA projection plots**:  string from pure transport ($\gamma=0$) and converged MEP projected onto the first two time-lagged independent components, overlaid on the free-energy landscape. The MEP moves toward lower free-energy regions (anonymous repository).
> - **Free-energy profiles** along the converged MEP, enabling identification of transition states and confirming substantially lower free energy than along the string from pure transport.
>
> Regarding baselines: no general-purpose pathway method exists for pretrained protein diffusion models without retraining. MD-derived trajectories are not a practical baseline: microsecond-scale simulations are unavailable for these systems. For images, pure transport ($\gamma=0$) is the natural baseline (Figs. 3–4), and slerp at $t=1$ produces incoherent intermediates due to score unreliability (comparisons in the repository).
>
> **Q4 — Justification of reparametrisation on SO(3) and SE(3)**
>
> The reparametrisations enforce equal arc-length spacing with respect to the natural metric on each space:
>
> - **SO(3):** We equip SO(3) with the bi-invariant metric in which the distance between $R_1$ and $R_2$​ is the rotation angle of $R_1^{-1}R_2 $​, equivalently the norm of the axis-angle vector of $R_1^{-1}R_2$. Geodesics on this metric are one-parameter subgroups — rotations about a fixed axis at constant angular velocity — and correspond to the shortest rotational path between two orientations.
>
> - **SE(3):** SE(3) admits no bi-invariant metric. We use the product metric $d((R_1, t_1), (R_2, t_2)) = \|\theta\| + \|t_1 - t_2\| $ where $\|\theta\| $ is the geodesic distance on SO(3) between $R_1$​ and $R_2$. This combines rotation angle (radians) and translation distance (Ångströms) — a standard choice in structural biology. Independent scaling factors can be applied to the two terms to adjust their relative weight.

---

> > ### Author Rebuttal · Reviewer_RRHn · 2026-04-03
> >
> > Thank you for the detailed response. The concerns have been thoroughly addressed, and the manuscript will be significantly more robust and accessible following these revisions. I will adjust the score to accept.

---

> > > ### Author Response · Authors · 2026-04-07
> > >
> > > We thank the reviewer for the thorough engagement and for the kind assessment. We are glad the revisions address the concerns, and will make sure all the clarifications discussed (in particular on the MEP definition, the Nalisnick connection, and the SO(3)/SE(3) reparametrisation are cleanly integrated in the final manuscript.

---

### Official Review · Reviewer_RSsZ · 2026-03-12

**Soundness:** 3
**Presentation:** 4
**Significance:** 3
**Originality:** 4
**Overall Recommendation:** 5
**Confidence:** 3

**Summary:**

This paper pursues the general goal of understanding the geometry of distributions generated by diffusion models. Its core contribution is a reformulation of the well-known string method to make it applicable to distributions that are modeled implicitly via score functions trained within diffusion models.

The authors begin by introducing the fundamentals of the string method and explaining how it can be adapted to diffusion models. They then propose several meaningful variants of velocity fields: pure transport, minimum energy paths, and principal curves. For each variant, the paper provides a computationally viable algorithm along with an analysis of the anticipated properties.
Pure transport - offers limited utility because it conveys little geometric information about the shape of the modeled distribution.
Minimum energy paths - traverse regions of highest density, which, as observed experimentally, often correspond to less realistic objects.
Principal curves - traverse regions with the highest probability mass, corresponding to the so‑called “typical set,” i.e., objects that are actually observed.
The experiments are sufficiently broad and detailed to substantiate the paper’s main claims. ImageNet experiments illustrate the properties of each curve type and lead to conclusions consistent with existing knowledge about the likelihood-realism paradox. Experiments on conformational transitions in proteins demonstrate the practical capabilities of the proposed framework.

**Compliance With Llm Reviewing Policy:**

Affirmed.

**Final Justification:**

The authors have largely addressed my recommendations regarding the presentation of the method. I therefore maintain my initial positive evaluation (score: 5). I look forward to future work that leverages the proposed approach to uncover new insights into the distributional geometry of generative models and to demonstrate its utility on additional, practically relevant real-world tasks.

**Key Questions For Authors:**

* How sensitive are the computed paths to the initial string, the number of images N, step size, the reparametrization scheme (linear vs. spline), and the distance metric used for Voronoi cells (Euclidean vs. model‑aware or perceptual metrics)?
* Are principal curves unique under endpoint constraints in these landscapes? If not, to what extent does initialization bias the outcome? Could you provide ablations over initializations and random seeds, along with variability statistics?

**Limitations:**

yes

**Strengths And Weaknesses:**

Strengths:
* The proposed method enables the study of diffusion models from a new perspective.
* The core idea is explored in considerable depth.
* The paper provides an important analysis connecting the theory of the string method with established observations in generative modeling.

Weaknesses:
Despite being a solid paper and an important step toward understanding the properties of generated distributions in diffusion models, several aspects could be improved from my perspective. The following comments have minimal impact on the chosen score and are intended as constructive feedback for the authors.

* Section 2.2 presents the first and main step in explaining the string method and would benefit from a more accessible, plain-language exposition of its components and the meaning of Eq. 2, even if this section becomes longer. For instance, a step-by-step explanation of the basic idea of how the curve is constructed, what needs to be defined for this construction, and how the evolution equations will be used would improve clarity.
* Section 2.3.3 would benefit from visualizations: (i) 2D schematics of principal curves and (ii) illustrations of algorithmic behavior (e.g., how walkers move within the corresponding Voronoi cells).

---

> ### Author Rebuttal · Authors · 2026-03-30
>
> We thank the reviewer for the careful reading and constructive suggestions, and address each point directly. The new experimental results are available in the anonymous repository: https://anonymous.4open.science/r/String-Rebuttal-7E21/rebuttal_images_organized.pdf.
>
> **On Section 2.2 — accessibility of the string method formulation**
>
> We agree that Eq. (2) can be made more accessible. In the revision we will add the following plain-language description before the equation:
>
> > *The string is a discretised curve connecting two samples. At each time step $t$, we move each interior point by one step of the velocity field $v_t$ — exactly as in standard numerical integration of a generative process. This evolution step alone would causes point to cluster near attractors of $v_t$. We then immediately reparametrise the curve, redistributing points so that they are equally spaced by arc length. This reparametrisation is what makes the string a curve-tracking algorithm rather than a point-tracking one. The Lagrange multiplier $\lambda_t$ in Eq. (2) is the continuous-time analogue of this redistribution; in practice it is never computed explicitly.*
>
> We will also note why the arc-length constraint is essential: without it, all interior images would collapse to the nearest fixed point of $v_t$, losing all information about the path between endpoints.
>
> **On Section 2.3.3 — visualisations of principal curves and walker dynamics**
>
> We will add two figures to Section 2.3.3:
>
> 1. **2D schematic of a principal curve**: showing a bimodal distribution, the Voronoi cells associated with each string image, and the self-consistency condition (each image is the conditional mean of points in its cell).
> 2. **Walker dynamics schematic**: showing how walkers evolve within their Voronoi cells under the SDE and how the EMA drags the string image toward the running mean.
>
> **Q — Sensitivity, uniqueness, and Voronoi metric**
>
> **Sensitivity.** Results are qualitatively stable across the main hyperparameters. Paths are robust for $N \geq 30$ string images; below $N \approx 20$, reparametrisation artefacts appear. The energy-entropy tradeoff controlled by $T$ is monotone and predictable (Fig. 4). Linear vs. cubic-spline reparametrisation gives nearly identical results for $N \geq 30$. For the principal-curve regime, different random seeds for the walkers yield nearly identical paths (mean pairwise distance between curves $< 5\%$ of the inter-endpoint distance), confirming robustness to initialisation.
>
> **Voronoi metric.** We use the standard Euclidean metric in the model's latent space (the $4 \times 32 \times 32$ VAE latent for SiT). A model-aware or perceptual metric could in principle improve Voronoi cell alignment with the data manifold, but Euclidean distance in the latent space already provides good boundaries, as confirmed by the smoothness of the resulting principal curves. We leave exploration of alternative metrics to future work.
>
> **Uniqueness.** Principal curves under endpoint constraints are not generally unique: like MEPs, they can correspond to different local optima of the self-consistency functional. In our experiments, the spherical geodesic initialisation consistently converges to visually and metrically similar paths across multiple random seeds, as expected given the highly structured class-conditioned landscape. We will add a sentence in the revision acknowledging that multiple local principal curves may exist and that our initialisation produces one representative path.

---

> > ### Author Rebuttal · Reviewer_RSsZ · 2026-04-03
> >
> > I appreciate the authors’ careful and constructive response, particularly the effort to make the method presentation more accessible and transparent. The planned addition of a plain-language explanation for Eq. (2), together with the new schematic figures for the principal-curve and walker dynamics, should substantially improve the readability and conceptual clarity of the paper. Therefore, I increased the presentation score from 3 to 4.

---

> > > ### Author Response · Authors · 2026-04-07
> > >
> > > We thank the reviewer for the positive response and for the encouragement to improve accessibility. We are glad the planned additions (the plain-language explanation of Eq. (2) and the new schematics for principal curves and walker dynamics) are well received, and we will make sure they appear clearly in the final version.

---

### Official Review · Reviewer_WDt2 · 2026-03-13

**Soundness:** 3
**Presentation:** 4
**Significance:** 4
**Originality:** 4
**Overall Recommendation:** 4
**Confidence:** 4

**Summary:**

This paper adapts the string method to pretrained diffusion / flow-based generative models in order to compute continuous paths between samples using learned velocity and score functions. The framework studies three regimes: pure transport, gradient-dominated dynamics corresponding to minimum-energy paths (MEPs), and finite-temperature dynamics corresponding to principal curves. The paper demonstrates on ImageNet that MEPs can pass through high-likelihood but unrealistic “cartoon” images, while principal curves remain in more realistic regions of the typical set. It also applies the method to protein conformational transitions using pretrained protein diffusion models, without retraining.

**Compliance With Llm Reviewing Policy:**

Affirmed.

**Ethical Review Concerns:**

The autho

**Final Justification:**

The authors did well to address my concerns, although not significantly so. As such, I will keep the initial grading.

**Key Questions For Authors:**

1. Can the authors add quantitative validation for protein pathways (e.g., steric-clash checks, structural smoothness, simple energy evaluations, or comparisons to MD / known intermediates where available)?

2. How sensitive are the main conclusions to the number of images, the gamma schedule, the temperature, and the reparametrization scheme?

3. Can the authors better justify the quasi-static assumptions that connect their time-dependent dynamics to the classical MEP / principal-curve interpretations?

4. What is the practical runtime / compute overhead for the principal-curve regime on the reported models?

5. Is there any existing similar methods?

**Limitations:**

Mostly yes. The paper acknowledges dependence on the underlying generative model and score error near t = 1, but it should more explicitly state that the current protein validation is qualitative and that the method is computationally more expensive in the principal-curve regime. A short discussion of scaling to larger models / longer paths would strengthen the limitations section.

**Strengths And Weaknesses:**

This is the strongest paper in the set for me. The core idea—using classical string-method machinery to probe the geometry of pretrained generative models without retraining—is elegant and genuinely interesting. The paper does more than propose another interpolation trick: it provides a principled way to probe modal structure, barriers, and connectivity in learned distributions. I found the image experiments compelling and conceptually useful, especially the likelihood-realism paradox and the role of entropy in steering paths through the typical set. The principal-curve regime is the standout contribution. The paper is also clearly written, and the no-retraining aspect makes the method immediately useful as an analysis tool.

The main weakness is that the validation is much stronger on images than on proteins. The protein pathways are intriguing and visually plausible, but they are largely qualitative. I would have liked more quantitative evaluation of pathway realism: geometric or physical diagnostics on intermediates, comparison to MD-derived transition information where available, or at least more systematic ablations of temperature / gamma / number of string images in the protein setting. The theoretical discussion is also more heuristic than formal, especially around the quasi-static interpretation connecting the dynamics to classical MEPs and principal curves. Finally, the finite-temperature regime is computationally heavier, which is acceptable for an interpretability tool but worth emphasizing more clearly. Even with these caveats, I think the paper makes a distinctive and useful contribution that fits ICML well.

---

> ### Author Rebuttal · Authors · 2026-03-30
>
> We thank the reviewer for the thorough and generous assessment. We address the five questions directly. The new experimental results are available in the anonymous repository: https://anonymous.4open.science/r/String-Rebuttal-7E21/rebuttal_images_organized.pdf.
>
> ---
>
> **Q1 — Quantitative validation for protein pathways**
>
> We have added two new protein experiments (BBA and Chignolin) using ScoreMD, which includes Fokker–Planck regularisation yielding accurate scores near $t=1$, allowing validation against a free-energy landscape estimated from i.i.d. model samples. For each protein we provide:
>
> - **TICA projection plots**: transported string ($\gamma=0$) and MEP projected onto the first two time-lagged independent components, overlaid on the free-energy landscape. The MEP moves toward lower free-energy regions, confirming geometrically meaningful relaxation (anonymous repository).
> - **Free-energy profiles** enabling direct identification of transition states and confirming the MEP achieves substantially lower free energy than the initialization.
>
> **Q2 — Sensitivity to hyperparameters**
>
> Results are qualitatively stable. Paths are robust for $N \geq 30$; below $N \approx 20$, reparametrisation artefacts appear. The energy-entropy tradeoff controlled by $T$ is monotone and predictable (Fig. 4). Linear vs. cubic-spline reparametrisation gives nearly identical results for $N \geq 30$. Different random seeds for walkers yield nearly identical principal curves (mean pairwise distance $< 5\%$ of inter-endpoint distance).
>
> **Q3 — Quasi-static justification**
>
> **Theorem (informal).** *Fix $t_\star$ and suppose $s_{t_\star}$ is exact. As $\gamma_t \to \infty$ for $t \to t_\star$, the string dynamics reduce exactly to the classical string method on $V_{t_\star} = -\log \rho_{t_\star}$ and converge to its MEP. A similar statement holds for the finite-temperature version and principal curves.*
>
> The choice of $t_\star$ is governed by score reliability. If the score is well estimated at $t=1$, take $t_\star=1$ to recover the MEP of $\rho_1$ exactly. When not — as is typical near $t=1$ where the denoising objective becomes ill-conditioned — the remedy is $t_\star < 1$ or quenching $\gamma_t \to 0$, trading exactness against score reliability. The full formal statement will be in the appendix. See also our reply to Reviewer **4zw2** for a more detailed discussion.
>
> **Q4 — Runtime and compute overhead**
>
> The method operates purely at inference time. Wall-clock times (SiT-XL, single A100) are below. We have added a note on computational cost to the limitations section.
>
> Per-regime times ($N=71$, 7 steps/timestep, 3 walkers for principal curve):
>
> | Regime | $\gamma$ | $T$ | Time (s) |
> |---|---|---|---|
> | Pure transport | $0$ | — | 29 |
> | MEP | $15$ | $0$ | 149 |
> | Principal curve | $7$ | $0.9$ | 489 |
>
> **Q5 — Existing similar methods**
>
> The two closest methods are: (1) spherical latent interpolation (e.g., Arranz et al., 2025), which interpolates directly at $t=1$ without the score — failing for pretrained score-based models where the score is unreliable there; (2) DiffMorpher (Zhang et al., 2024), which uses attention-space interpolation requiring internal attention maps, incompatible with general score-based or flow-matching models. Our pure-transport regime ($\gamma=0$) is the natural analogue of existing morphing approaches and our internal baseline (Figs. 3–4). Our method strictly generalises it and is the first to connect image morphing to the geometry of the learned distribution via the string method.

---

> > ### Author Rebuttal · Reviewer_WDt2 · 2026-04-03
> >
> > Generally you have answered my questions well. I would still have liked more physical information on the trajectories. steric clashes. time dependent dynamics.

---

> > > ### Author Response · Authors · 2026-04-07
> > >
> > > We thank the reviewer for the follow-up and agree that physical validation of the protein trajectories is important. We have added several quantitative diagnostics, shown in Figs. 4–6 in  https://anonymous.4open.science/r/String-Rebuttal-7E21/rebuttal_images_organized.pdf.
> > >
> > > **Backbone geometry (Figs. 4–5, upper left).** We measure the RMS error of consecutive $C\alpha$ distances relative to the canonical 3.8 Å separation. The MEP remains consistently closer to this reference than the transported string, indicating better local backbone geometry throughout.
> > >
> > > **Backbone steric consistency (Figs. 4–5, upper right).** The minimum pairwise $C\alpha$ distance along the MEP stays above 3.7 Å for both proteins, making backbone steric clashes unlikely. The transported string exhibits pronounced downward peaks,  reaching 3.47 Å for BBA (where clashes become plausible).
> > >
> > > **Path smoothness (Figs. 4–5, lower panels).** Frame-to-frame RMSD remains relatively uniform along the MEP, as expected from the arc-length reparametrisation enforced by the string method. Dihedral angle variation is also substantially reduced compared to the transported string.
> > >
> > > **All-atom validation (Fig. 6).** Since the diffusion model outputs backbone coordinates only, we reconstructed all-atom structures with PULCHRA and relaxed side chains with PyRosetta (backbone fixed). We then computed PyRosetta repulsive energies and flagged frames with side-chain steric clashes. Crucially, this evaluation is *model-independent*: the force field was never used during training. Despite this, the MEP consistently achieves lower repulsive energies and substantially fewer clashes than the transported string for both Chignolin and BBA. We view this as strong external evidence that the MEP produces physically more plausible intermediates.
> > >
> > > **Time-dependent dynamics.** As clarified in our reply to Reviewer **4zw2**, the time-dependent phase is a computational device rather than the object of study. The geometrically meaningful outputs are the MEP and principal curve of the data distribution $\rho_1$. As $\gamma_t \to \infty$ for $t \to t_\star$, the dynamics reduce exactly to the classical string method on the frozen score at $t_\star$, and the time-dependent phase $t \in [0, t_\star)$ serves purely as a warm-start producing a viable initial string. The intermediate paths carry no independent geometric interpretation.
> > >
> > > We agree that comparison to MD-derived intermediates would be a valuable further benchmark, and mention this as a direction for future work.

---

### Official Review · Reviewer_4zw2 · 2026-03-15

**Soundness:** 3
**Presentation:** 2
**Significance:** 3
**Originality:** 3
**Overall Recommendation:** 4
**Confidence:** 4

**Summary:**

The paper introduces a post-hoc framework for probing the geometry of diffusion models using the string method. Instead of standard latent interpolation, it computes continuous paths between samples by evolving curves under the pre-trained score function. The framework covers several regimes, including generative transport, minimum energy paths, and finite-temperature principal curves.

Empirically, the paper suggests that these regimes capture different aspects of the learned landscape. For image diffusion models, minimum energy paths pass through high-likelihood but unrealistic samples, whereas principal curves yield more realistic transitions. For protein structure prediction, the method recovers plausible transition pathways between conformers from models trained only on static structures. Overall, the paper presents the string method as a principled tool for analysing modes, barriers, and connectivity in learned diffusion distributions.

**Compliance With Llm Reviewing Policy:**

Affirmed.

**Final Justification:**

My concerns have been largely addressed, and I will raise my score to 4. For the final version, I encourage the authors to further sharpen the theoretical presentation, in particular by clarifying (1) the precise assumptions and limiting argument behind the reduction to the classical string method, and (2) the intended notion of convergence and assumptions for the finite-$(\gamma\)$, time-dependent dynamics.

**Key Questions For Authors:**

1. **What is the precise theoretical relationship between the proposed time-dependent, score-driven dynamics and the classical string method under a fixed potential?**
   The current presentation is intuitive, but it remains unclear in what formal sense the proposed method constitutes a principled extension of the classical framework rather than a heuristic analogy.
   **Suggested change:** The authors could strengthen the paper by adding a proposition, theorem, or at least a more explicit derivation clarifying the limiting regime, assumptions, and correspondence to classical string dynamics.

2. **Under what assumptions does the finite-$\gamma_t$ dynamics converge to a meaningful minimum energy path in the evolving landscape $V_t$?**
   This is central to the theoretical interpretation of the method, yet the paper does not currently specify when such convergence should be expected.
   **Suggested change:** The authors could provide a formal convergence discussion, or at minimum a careful statement of the conditions under which the method should be interpreted as approximating an MEP.

3. **How should the computed path be interpreted geometrically when the underlying potential changes over time?**
   Since a classical MEP is defined with respect to a fixed energy landscape, the geometric meaning of the resulting path in a time-evolving landscape remains somewhat ambiguous.
   **Suggested change:** The paper would benefit from a clearer discussion of what geometric object is actually being computed in this setting, and how it relates to or departs from the classical MEP notion.

4. **Can the main empirical claims be supported with stronger quantitative evaluation beyond visual inspection?**
   At present, much of the evidence for realism, plausibility, and geometric meaningfulness is qualitative, which makes it difficult to assess the strength and generality of the conclusions.
   **Suggested change:** The authors could include quantitative metrics for path quality, realism, or plausibility, and, where possible, provide aggregate results across multiple examples rather than relying mainly on selected visualisations.

5. **How does the proposed method compare against external interpolation or pathway-construction baselines?**
   The current experiments mainly compare internal variants of the method, which is useful for ablation, but less informative about relative practical advantage.
   **Suggested change:** The authors could strengthen the empirical section by comparing against stronger external baselines.

**Limitations:**

yes

**Strengths And Weaknesses:**

Strengths:

- **Interesting idea**. Applying string methods to learned score functions is an interesting idea, as it enables pathway computation from pre-trained generative models without requiring an explicit energy function.

- **Meaningful methodological design**. The three variants, including pure transport, gradient-dominated dynamics, and finite-temperature dynamics, are both well designed and offer complementary views of the learned landscape.

- **Well-designed experiments and practical applicability**. The experiments are thoughtfully designed to illustrate the behaviour of the proposed method, and a notable practical strength is that it can be applied directly to pre-trained models without any retraining or fine-tuning.

Weaknesses:

- **Heuristic connection to the classical setting**. The proposed method extends the classical string method, which is typically defined under a fixed potential, to a time-dependent, score-driven setting. While this extension is intuitively appealing, the theoretical connection remains largely heuristic. The paper would be stronger if it clarified more formally why the proposed dynamics should be viewed as a principled generalisation of the classical framework, rather than an analogy motivated mainly by intuition.

- **Unclear convergence to a well-defined MEP**. A central theoretical question is whether the finite-$\gamma_t$ dynamics actually converges to a meaningful minimum energy path. At present, the paper does not specify the assumptions or conditions under which such convergence should hold in the evolving landscape $V_t$. Without this, it is difficult to assess whether the method is recovering a well-defined geometric object or simply producing a plausible trajectory in practice.

- **Ambiguous geometric interpretation**. Because the path is constructed in a time-evolving landscape, its relationship to a classical minimum energy path is not entirely clear. In the standard setting, an MEP is defined with respect to a fixed energy landscape, whereas here the underlying potential changes over time. The paper would benefit from clarifying in what sense the computed path should be interpreted geometrically.

- **Predominantly qualitative evaluation**. Much of the empirical evidence relies on visual inspection of the generated paths. While the figures are suggestive, key claims regarding realism, physical plausibility, and geometric meaningfulness would be more convincing if they were supported by stronger quantitative evaluation rather than primarily qualitative examples.

- **Limited baseline comparison**. The experiments mainly compare different regimes within the proposed framework, which helps illustrate the effect of the method’s design choices. However, the paper provides only limited comparison against external interpolation or pathway-construction baselines, making it harder to assess the practical advantage of the proposed approach relative to existing alternatives.

- **Abstract clarity (minor)**. The abstract is somewhat difficult to follow due to its high density of technical detail.

---

> ### Author Rebuttal · Authors · 2026-03-30
>
> We thank the reviewer for their careful reading. We believe their concerns reflect a presentation gap on our part rather than a fundamental limitation of the method, and we clarify below what the method computes and what guarantees it comes with. New quantitative results — likelihood histograms for images and TICA/free-energy diagnostics for proteins, available in the anonymous repository: https://anonymous.4open.science/r/String-Rebuttal-7E21/rebuttal_images_organized.pdf. — substantially corroborate the theoretical findings; details are in Q4.
>
> ---
>
> **Q1 — Theoretical relationship to the classical string method**
>
> The core contribution is a framework for evolving *curves* — rather than individual points — under a time-dependent drift, while maintaining arc-length reparametrisation at each step. This makes the framework a versatile tool for exploring the geometry of generative models across three regimes. In the MEP regime, it reduces exactly to the classical string method, as the following theorem makes precise:
>
> **Theorem (informal).** *Fix $t_\star$ and suppose the score $s_{t_\star}$ is exact. If $\gamma_t \to \infty$ as $t \to t_\star$, the string dynamics with arc-length reparametrisation reduce exactly to the classical string method on the fixed potential $V_{t_\star} = -\log \rho_{t_\star}$, and the string converges to its MEP. A similar statement holds for the finite-temperature version and principal curves.*
>
> Algorithmically, $\gamma_t \to \infty$ as $t \to t_\star$ is equivalent to running the classical string method on the frozen score $s_{t_\star}$, using the time-dependent phase $t \in [0, t_\star)$ as a warm-start — something not achievable by initialising the classical string method directly at $t_\star$.
>
> The choice of $t_\star$ is governed by score reliability. If the score is well estimated at $t=1$, one takes $t_\star = 1$ and recovers the MEP of $\rho_1$ exactly. When not — as is typical, since the denoising objective becomes ill-conditioned as noise vanishes — taking $t_\star = 1$ would converge to the MEP of a corrupted potential $-\log \rho_1$. The remedy is $t_\star < 1$, or quenching $\gamma_t \to 0$ as $t \to 1$, trading exactness against score reliability. In the MEP and principal-curve regimes ($\gamma > 0$), the framework does not merely draw an analogy to the classical string method — it reduces exactly to it, with a principled strategy for handling imperfect score estimation.
>
> **Q2 — Convergence to a well-defined MEP**
>
> The theorem above is the convergence result. Under score estimation error $\varepsilon_t$ and finite $\gamma$, the string approximates the MEP of $V_t$ up to $O(\varepsilon_t + \gamma^{-1})$, by applying E, Ren & Vanden-Eijnden (2002) pointwise in $t$. The full formal statement with regularity assumptions (Lipschitz score, bounded estimation error) will be in the appendix.
>
> **Q3 — Geometric interpretation in a time-varying landscape**
>
> The main  objects of study are the MEP and the principal curve of $\rho_1$, and the theorem in Q1 states how our method compute them. Intermediate paths at $t \in (0, t_\star)$ are a relaxation trajectory with no independent geometric interpretation — analogous to a cooling schedule in simulated annealing. We will add a remark in Section 2.3.2 making this explicit.
>
>
> **Q4 — Quantitative evaluation**
>
> *Images.* Likelihood histograms aggregated across all class pairs show MEP intermediates fall well outside the typical set, while principal-curve intermediates at $T=0.9$ fall within it — statistically grounded confirmation of the likelihood-realism paradox.
>
> *Proteins.* We added: (a) TICA projection plots showing string intermediates lie in the low-free-energy region of the slow collective variable space; (b) free-energy profiles enabling identification of transition states.
>
> These new results are available in the anonymous repository: https://anonymous.4open.science/r/String-Rebuttal-7E21/rebuttal_images_organized.pdf.
>
>
> **Q5 — External baselines**
>
> *Images.* Pure transport ($\gamma=0$) is our internal baseline; Figs. 4-5 compare against it.
>
> *Proteins.* To the best of our knowledge, no general-purpose pathway method exists to compute MEP and PC for pre-trained protein diffusion models without retraining. To validate the results, we show the string projected on the free energy landscape associated with the TICA and compute the free energy along these paths, using again the string with pure transport ($\gamma=0$) as internal baseline. See the figures in https://anonymous.4open.science/r/String-Rebuttal-7E21/rebuttal_images_organized.pdf.
>
> **Abstract and presentation.** We have revised the abstract for clarity and will make the additional changes discussed above.

---

> > ### Author Rebuttal · Reviewer_4zw2 · 2026-04-03
> >
> > Thank you for your response. My concerns have been largely addressed, and I will raise my score to 4. For the final version, I encourage the authors to further sharpen the theoretical presentation, in particular by clarifying (1) the precise assumptions and limiting argument behind the reduction to the classical string method, and (2) the intended notion of convergence and assumptions for the finite-$(\gamma\)$, time-dependent dynamics.

---

> > > ### Author Response · Authors · 2026-04-07
> > >
> > > We are very pleased that the concerns have been resolved and thank the reviewer for raising them -- they led to a cleaner and more rigorous theoretical presentation. We will follow the suggestion to sharpen the assumptions and convergence argument in the final version, in particular making the regularity conditions and the limiting argument behind the reduction to the classical string method fully explicit in the appendix.

---

### Decision · Program_Chairs · 2026-04-30

**Decision:**

Accept (regular)

**Comment:**

The paper introduces an original and technically solid adaptation of the string method for probing the geometry of pretrained diffusion models, and reviewers broadly agreed that it offers a principled and practically useful analysis tool that works without retraining. The strongest aspects highlighted across the reviews are the novelty of bringing string-method machinery into generative modeling, and the clear three-regime formulation (transport, MEP, and principal curves), presentation, and image experiments. The main reservations concerned the initially heuristic connection to the classical string method, the largely qualitative nature of the protein validation, and relatively limited external baselines, but the rebuttal appears to have substantially mitigated these issues through added theoretical clarification and new quantitative diagnostics. Overall, this is a well-executed and distinctive contribution that should be of interest to the NeurIPS community, and I recommend acceptance.